# Trends and predictors of in-hospital mortality among babies with hypoxic ischaemic encephalopathy at a tertiary hospital in Nigeria: A retrospective cohort study

Beatrice Nkolika Ezenwa[1,2]*, Gbenga Olorunfemi[3,4], Iretiola Fajolu[1,2], Toyin Adeniyi[2], Khadijah Oleolo-Ayodeji[2], Blessing Kene-Udemezue[2], Joseph A. Olamijulo[3], Chinyere Ezeaka[1,2]

1 Neonatology Unit, Department of Paediatrics, College of Medicine University of Lagos, Lagos, Nigeria, 2 Department of Paediatrics, Lagos University Teaching Hospital, Lagos, Nigeria, 3 Department of Obstetrics & Gynecology, Lagos University Teaching Hospital, Lagos, Nigeria, 4 Division of Epidemiology and Biostatistics, School of Public Health, University of Witwatersrand, Johannesburg, South Africa

* beatriceezenwa@yahoo.com

## Abstract

### Background

Globally, approximately 9 million neonates develop perinatal asphyxia annually of which about 1.2 million die. Majority of the morbidity and mortality occur in Low and middle-income countries. However, little is known about the current trend in incidence, and the factors affecting mortality from hypoxic ischaemic encephalopathy (HIE), in Nigeria.

### Objective

We assessed the trends in incidence and fatality rates and evaluated the predictors of mortality among babies admitted with HIE over five years at the Lagos University Teaching Hospital.

### Methods

A temporal trend analysis and retrospective cohort study of HIE affected babies admitted to the neonatal unit of a Nigerian Teaching Hospital was conducted. The socio-demographic and clinical characteristics of the babies and their mothers were extracted from the neonatal unit records. Kaplan-Meir plots and Multivariable Cox proportional hazard ratio was used to evaluate the survival experienced using Stata version 16 (StataCorp USA) statistical software.

### Results

The median age of the newborns at admission was 26.5 (10–53.5) hours and the male to female ratio was 2.1:1. About one-fifth (20.8%) and nearly half (47.8%) were admitted within 6 hours and 24 hours of life respectively, while majority (84%) of the infants were out-born. The prevalence and fatality rate of HIE in our study was 7.1% and 25.3% respectively. The

**Data Availability Statement:** All relevant data are within the manuscript and its Supporting information files.

**Funding:** The authors received no specific funding for this work.

**Competing interests:** The authors have declared that no competing interests exist.

annual incidence of HIE among the hospital admissions declined by 1.4% per annum while the annual fatality rate increased by 10.3% per annum from 2015 to 2019. About 15.7% died within 24 hours of admission. The hazard of death was related to the severity of HIE (p = 0.001), antenatal booking status of the mother (p = 0.01) and place of delivery (p = 0.03).

## Conclusion

The case fatality rate of HIE is high and increasing at our centre and mainly driven by the pattern of admission of HIE cases among outborn babies. Thus, community level interventions including skilled birth attendants at delivery, newborn resuscitation trainings for healthcare personnel and capacity building for specialized care should be intensified to reduce the burden of HIE.

## Introduction

Hypoxic ischaemic encephalopathy (HIE) is a neurological complication from the inability to establish and sustain respiration at birth in a newborn. It is the commonest cause of neonatal encephalopathy [1] and a major cause of neonatal morbidity and mortality globally [1–4]. HIE associated deaths are the fifth most common cause of under-five deaths accounting for 814,000 deaths annually [3, 5]. Several studies have documented on the short and long term adverse effects of HIE in infants such as increased requirements for supportive care in the perinatal period, severe long-lasting neurological sequelae, cerebral palsy, epilepsy and cognitive impairments [6–8]. In most high-income countries, the incidence of HIE has reduced significantly following improvements in obstetric and neonatal care unlike in lower and middle-income countries (LMICs) [8, 9]. In Sub-Saharan Africa and indeed in Nigeria, the real burden of HIE is difficult to determine due to the paucity of data [10]. Aliyu et al [3] reported that HIE contributed as high as 42 million disability-adjusted life years among affected individuals. Newborns that suffer HIE have higher mortality rates and it contributes to high rates of morbidity in neonatal survivors [11, 12]. Several factors influence the survival rate in babies that developed HIE. Such factors include the place of birth, level of prenatal care, the cause of asphyxia, gestational age, maternal age, maternal illness, socioeconomic status, availability of resources for neonatal care and time to get to specialized care [4, 6]. Furthermore, the grade of HIE can also impact on survival. Thus, the Sarnat and Sarnat staging of HIE has been identified as a predictive tool in prognosticating severe perinatal asphyxia. However, this scoring tool requires a certain level of medical expertise [9, 10, 13, 14].

Over the last decade, Nigeria embarked on a nationwide training of healthcare providers on strategies to eliminate preventable newborn deaths [15, 16]. The emphasis was majorly on prevention strategies. Health care providers were mandated and encouraged to undertake the newborn resuscitation and helping babies breathe trainings while mothers were encouraged to register and attend antenatal care and also deliver at facilities with skilled birth attendants [17]. The trainings empowered the health care providers in the peripheral centers to promptly identify and refer cases of HIE that will require tertiary level care. These strategies were expected to reduce perinatal morbidity and mortality and improve newborn survival in Nigeria. There is no recent evidence of the impact of these interventions on the trends in HIE in our environment to guide the Government on how to update and improve on current public health intervention strategies. A hospital-based study on the burden of HIE can assist in prioritizing personnel and equipment to effectively manage the HIE cases that may present to the hospital. While previous researchers in Nigeria had utilized Cox proportional analysis to evaluate

neonatal mortality, appropriate robust cohort analysis were not utilized for analysis of in-hospital mortality of babies with HIE. Therefore, we aimed to evaluate the trends and the predictors of immediate in-hospital mortality among babies with HIE at a tertiary care hospital in Nigeria during a five-year period to inform further interventions.

## Materials and methods

This study was a hospital-based temporal trends analysis and retrospective cohort study of babies admitted to the neonatal units of the Lagos University Teaching Hospital (LUTH) from 1 January 2015 to 31st December, 2019. LUTH is one of the three publicly funded tertiary health institutions in Lagos, South-Western Nigeria. Lagos is a cosmopolitan City with an estimated population of over 14 million residents according to the World Population Review [18]. LUTH also serves the neighboring States and receives referral from primary and secondary health facilities including private health facilities. The neonatal unit of the hospital had 80 cot/ incubator spaces with inborn and outborn sections. The inborn neonatal ward adjoined the labour ward and all caesarean sections and high-risk deliveries were attended by the Paediatricians and/ or the Neonatologists. The neonatal unit had no functional mechanical ventilator but both conventional and improvised bubble Continuous Positive Airway Pressure (CPAP) devices were available for infants requiring respiratory support. The hospital laboratories and imaging units also serve the neonatal wards though certain specialized investigations such as arterial blood gases and magnetic resonance imaging studies were not routinely done. The admitted newborns with HIE were managed according to the unit protocol for HIE which was majorly supportive care [14] with average nurse-patient ratio of 1:6. The socio-demographic characteristics, birth history and clinical characteristics of the babies and their mothers were extracted from the neonatal unit admission records into an excel spreadsheet for analysis. Only infants whose primary admission diagnosis was HIE were included in the study. HIE was defined for the purposes of this study as the presence of encephalopathy or altered consciousness and multi-organ failure in a term newborn with a positive history of delayed cry at birth or required prolonged resuscitation at birth in addition to the presence of any of the neurological features as contained in the Sarnat and Sarnat classification [12, 13]. Infants with gross congenital anomalies or other primary diagnosis such as sepsis with fever at presentation were excluded.

Data collected included: maternal age, parity, mode of delivery, age of infant at admission, sex, gestational age, place of delivery (inborn/outborn), birth weight, duration of admission and outcome (death, or survival). Babies whose mothers delivered in LUTH were classified as "inborn". Mothers who delivered in LUTH but did not receive antenatal care in LUTH were further classified as "inborn unbooked" to distinguish them from "inborn booked" patients who were patients that their mothers had antenatal care in LUTH and delivered in LUTH. Those babies that were delivered outside LUTH were classified as "Outborn". The "Outborn" babies were delivered either at the government health care facilities, private health facilities, facilities run by Traditional birth attendants (TBA) or at home before referral to LUTH. These referred infants were usually very ill and required intensive care. As per the protocol of the unit, on the admission of each baby, the severity of HIE was determined by the senior registrar, consultant paediatrician or the neonatologist that first examined the baby based on the Sarnat and Sarnat staging. The HIE severity was classified as mild (Sarnat stage 1), moderate (Sarnat stage 2) and severe (Sarnat stage 3) [13, 14].

### Ethical considerations

This is a retrospective anonymous data collection with no potential ethical breach. However, the data collection process of the database that was utilized has ethical approval from the Health Research and Ethics Committee of LUTH with approval number ADM/DCST/HREC/APP/1465.

## Statistical analysis

The data was imported into Stata version 16 (StataCorp, Texas USA) statistical software for analysis from excel spreadsheet. Data cleaning and validation was done. Categorical variables were described as frequencies and percentages while continuous variables were presented as mean (± standard deviation) or median (interquartile range)–if not normally distributed.

The annual incidence of HIE cases seen in our hospital was calculated by dividing the annual number of HIE cases by the number of neonatal admissions. The annual case fatality was also calculated by dividing the number of HIE deaths by the annual number of HIE cases multiplied by 100. The incidence and fatality rate were then plotted on graphs and the annual percent change of the trends were calculated from 2015 to 2019. The prevalence (with 95% Confidence interval) of HIE was also calculated. The prevalence (with 95% Confidence interval) of the grades or severity of HIE was also calculated. The association between neonatal and maternal characteristics and severity of HIE was assessed using Pearson's Chi-square or Fischer's exact (for categorical variables), one-way analysis of variance (for continuous variables) or Kruskal Wallis test (for non-normally distributed continuous variables).

Time to death was the time-varying variable. Babies that died were coded as 1 while babies that were discharged alive were right-censored and coded as 0. Life tables of the survival experience was produced. Survival experience by various baseline categorical characteristics (sex, severity, inborn/outborn etc) were compared using the Kaplan Meir survival plots and log-rank test. Univariable and multivariable Cox proportional hazard regression modelling was performed with the baseline characteristics as the covariates. Variables with univariable p-value < 0.2 were used to build the multivariable model using backward elimination method. Some variables were selected *a priori*. Crude and adjusted hazard ratio (95% confidence interval) were calculated. Final multivariable Cox proportional hazard model adjusted for HIE severity, gender, weight, age at presentation, type of facility, year of admission. Some variables such as baby's age, birth weight was handled both as continuous and categorical variable. Two-tailed test of hypothesis was assumed and a P-value <0.5 was set as statistically significant level. Post regression estimation tests such as the Schoenfeld's test of proportional hazard assumptions was conducted and a p-value >0.05 showed no violation of the assumption of proportional hazard. The variance inflation factor was conducted and a value < 10 shows no collinearity among explanatory variables.

## Results

### Socio demographic characteristics

During the 5-year study period spanning January 2015- December 2019, a total of 4399 ill infants were admitted, into the neonatal wards of LUTH. Among these, 312 babies fulfilled the Sarnat and Sarnat criteria for HIE. Table 1 shows the Socio-biologic characteristics of infants with HIE. The median age at admission of the infants that suffered HIE was 27 (10–53) hours while the youngest and oldest baby at admission were within 1st hour of life and 144 hours (6 days) old, respectively. About one-fifth of the babies (n = 65/312, 20.8%), comprising all the inborn babies and only 11 outborn infants, were admitted within 6 hours of life while nearly half of the babies were admitted within 24 hours of life n = 149/312, 47.8. The male to female ratio was 2.1:1. Majority, (n = 257/312, 84%) of the admitted babies were outborn; and three-fifth, (n = 30/49, 61.22%) of the 49 inborn babies, were delivered by unbooked mothers in LUTH. Almost all (n = 296/312, 94.9%) the infants were delivered at term (≥37 weeks) and median weight at admission was 3000g (IQR: 2700–3475) (Table 1).

**Table 1. Socio-biologic characteristics of infants with HIE.**

| Variable | Frequency (%) |
|---|---|
| **Age of neonate (hours) (median, IQR)** | 27 (10–53) |
| ≤6hours | 65 (20.8) |
| 7–24 | 84 (26.9) |
| 25–72 | 153 (49.0) |
| >72 | 10 (3.2) |
| **Gender** | |
| Female | 102 (32.7) |
| Male | 210 (67.3) |
| **Gestational age at delivery (weeks)** | |
| Pre-term (35–37) | 14 (4.5) |
| Term (37–41) | 296 (94.9) |
| Post-term (>41) | 2(0.6) |
| **Weight (gram) (median, IQR)** | 3000 (2700–3475) |
| <2500 | 36 (11.5) |
| 2,500–3999 | 258 (82.69) |
| ≥4000 | 18(5.77) |
| **Place of birth** | |
| Inborn (Booked) | 19 (6.2) |
| Inborn (Unbooked) | 30 (9.8) |
| Outborn | 257 (84.0) |
| Not stated | 6 (1.9) |

## Maternal characteristics

Information was only available for the inborn mothers with 47 out of the 49 mothers having complete data. The mean maternal age and median parity was 28.02± 5.33 years and 2 (1–3) respectively. Only 6.2% of the mothers booked in LUTH and about four-fifth (n = 38/47, 80.85%) of the mothers that delivered in LUTH had an emergency caesarean section (Table 2).

## Prevalence of HIE, pattern of presentation and outcome

The prevalence of HIE among newborn admissions during the study period was 7.1% (312/4399). The majority of the infants presented with moderate HIE (n = 190/312, 60.9%, 95% CI:55.34%– 66.19%), followed by severe HIE (n = 75/312, 24.0% (95%CI:19.60%– 29.12%). Mild cases were the least common (n = 47/312, 15.1% (95%CI: 11.49%–19.50%). Majority (n = 224/312, 71.79%, 95%CI:) of the babies were discharged while the case fatality rate was 25.32% (95%CI: 20.78%– 30.47%, n = 79/312,).

## Trends in the annual incidence, mortality, fatality rate and severity of HIE (2015–2019)

The number of HIE admission decreased from 63 admissions in 2015 to 46 admissions in 2018 and then increased to 86 admissions in 2019 (Fig 1A, S1 Table). Similarly, there was a decrease in the annual HIE incidence from 9.17 per 100 neonatal admissions in 2015 to 5.19 per 100 neonatal admissions in 2017 at a rate of 18.4% per annum. Subsequently, there was an increase in incidence from 5.19 per 100 admissions in 2017 to 8.54 per 100 admissions in 2019 at a rate of 74.0% per annum. The overall HIE incidence from 2015 (9.17 per 100 neonatal admissions) to 2019 (8.54 per 100 admissions) showed a decline of 1.4% per annum (Fig 1B, S1 Table).

**Table 2. Demographic and obstetrics characteristics of the mothers of inborn infants with HIE.**

| Maternal characteristics, n = 49 | Frequency (%) |
|---|---|
| **Booking status of mothers, n = 49** | |
| Booked | 19 (6.2) |
| Unbooked | 30 (9.8) |
| Not stated | 6 (1.9) |
| **Maternal age (years), n = 47 Mean ± SD** | 28.0± 5.3 |
| <25 | 15(31.9) |
| 25–29 | 8 (17.0) |
| 30–34 | 19 (40.4) |
| 35–37 | 5 (10.6) |
| **Parity (median, IQR), n = 47** | 2(1–3) |
| 1 | 21 (44.7) |
| 2 | 12 (25.5) |
| 3 | 6 (12.8) |
| 4 | 7 (14.9) |
| 5 | 0(0.0) |
| 6 | 1 (2.1) |
| **Mode of delivery, n = 47** | |
| Vaginal delivery | 7(14.9) |
| Assisted Vaginal delivery | 2 (4.6) |
| Emergency Caesarean section | 38 (80.9) |

The annual fatality rate increased from 17.46% in 2015 to 30.23% in 2019 at a rate of 10.3% per annum (Fig 1B, S1 Table). Both moderate and severe HIE increased in proportion from 2015 to 2017 then slightly decline in proportion while mild cases decreased from 2015 to 2017 and then slightly increased till 2019 (Fig 1C).

From Table 3, higher proportion of babies with normal birth weight (2,500–3999 gram) were admitted for HIE during the time-period 2015–2017 as compared to the time-period 2018–2019 (2015–2017 vs 2018–2019: 76.7% vs 88.6, P-value 0.024). However, there was no statistically significant association between the other maternal and babies' characteristics and the period of admission.

## Association between severity of HIE and socio biological characteristics

From Table 4, the median duration of hospital stay was shortest among the babies with severe HIE, 96 (IQR: 24–240) hours as compared to babies with mild HIE, 240 (IQR: 144–240) hours, and moderate HIE, 240 (IQR: 168–312) hours, respectively (P-value = 0.0001). Furthermore, babies with mild HIE had higher median birth weight (3200 (IQR: 2800–3600) g) as compared to babies with moderate 3000 (IQR: 2700–3400) or severe HIE 3000 (IQR: 2500–3400) grams. Although not reaching a statistically significant level. (P-value = 0.075).

## Association between survival of HIE affected babies and maternal/neonatal characteristics

The prevalence of death increased with increasing severity of HIE and the prevalence of death was about three to four fold among babies with severe cases (53.3%, (95%CI: 41.8%–64.5%, n = 40/75) as compared to babies with mild (12.8% (95%CI: 5.7%–26.3%, n = 6/47), or moderate HIE cases (17.4% (95%CI: 12.6%–23.5%, n = 33/190) (P-value < 0.0001). Babies that

(A)

(B)

(C)

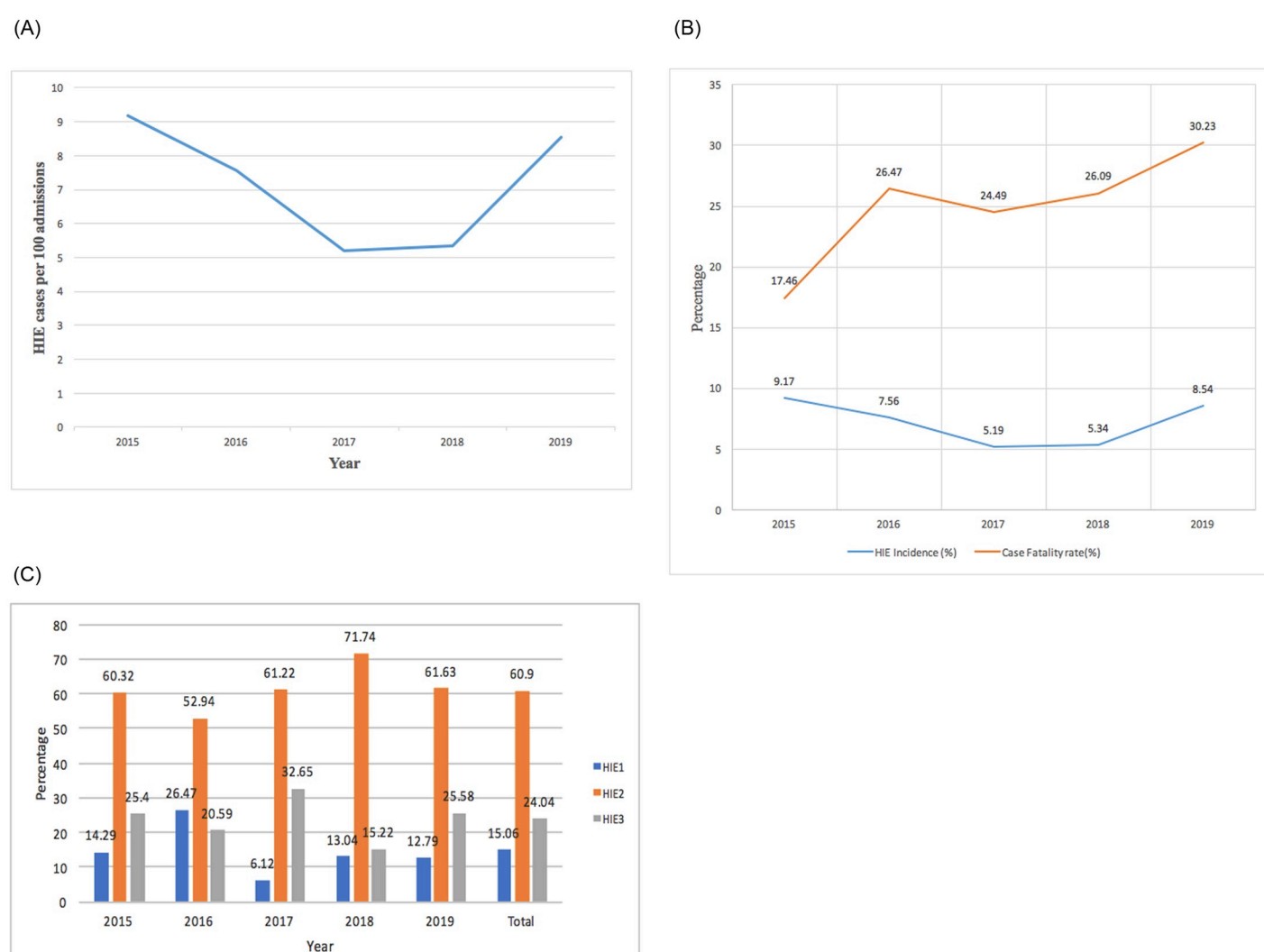

**Fig 1.** A. Trends in HIE admission. B. Trends in incidence and fatality rate for HIE (2015–2019). C. Trends in severity of admitted cases of HIE (2015–2019).

survived had longer hospital stay than those that died. 24 (24–48) hours Vs 240(216–336) hours, P-value < 0.001. Out of the 47 babies with mild HIE, 6 (12.8%) died, while 33 out of the 190 (17.4%) and 40 out of the 75(53.3%) babies with moderate and severe HIE respectively, died. (Table 5) All the babies with mild HIE that died were outborn infants and all except two presented to the facility after 24 hours of life.

**Survival experience of the babies with HIE.** The overall survival experience of the cohort is as shown in Fig 2(A). The babies contributed 2743.08 neonate-days of follow-up and about one-quarter (25.3% (n = 79 /312) died. The death rate was 28.8 per 1000 neonate-days. About 14.7% (n = 46/312) died within the first 24 hours of admission while 22.1% (n = 69/312) died within 72 hours. The cumulative risk of survival at 1st, 3rd, 7th and 14th day of admission was 85.8%, (95%CI:80.8%–88.7%), 78.4% (95% CI: 72.8%–82.1%), 75.6%, (95%CI: 69.9%–79.6%) and 74.7%, (95%CI: 69.0%–78.8%) respectively (S2 Table).

Fig 2(B) showed the survival experience of the babies based on severity of disease. As expected, babies with severe disease died faster than the others with 75% and 50% of severe

Table 3. Maternal and neonatal characteristics by time period.

| Characteristics | 2015–2017 N = 180 (%) | 2018–2019 N = 132 (%) | P-value[a] |
|---|---|---|---|
| **Severity of HIE** | | | |
| Mild (HE1) | 30 (16.7) | 17(12.9) | 0.4 |
| Moderate (HIE2) | 104 (57.8) | 86 (65.2) | |
| Severe (HE3) | 46 (25.6) | 29 (22.0) | |
| **Age of neonate at presentation(hours) (median, IQR)** | 34(11–72) | 24 (10–48) | 0.09[b] |
| 1-6hrs | 37 (20.6) | 28 (21.2) | 0.4 |
| 7–24 | 44 (24.4) | 40 (30.3) | |
| 25–72 | 91 (50.6) | 62 (47.0) | |
| ≥72 | 8 (4.4) | 2 (1.5) | |
| **Gender** | | | |
| Female | 66 (36.7) | 36 (27.3) | 0.1 |
| Male | 114 (63.3) | 96 (72.7) | |
| **Gestational age at delivery** | | | |
| Term | 170 (94.4) | 128 (97.0) | 0.3 |
| Preterm | 10 (5.6) | 4 (3.0) | |
| **Birth Weight (gram) (median, IQR)** | 3000 (2750–3500) | 3000 (2600–3400) | 0.2 |
| <2500 | 28 (15.6) | 11(8.3) | 0.02 |
| 2,500–3999 | 138 (76.7) | 117 (88.6) | |
| ≥ 4000 | 14 (7.8) | 4 (3.0) | |
| **Maternal age (years) Mean ± SD, (N = 47)** | 28.0 ± 5.6 | 28.1 ± 5.4 | 0.97[c] |
| <25 | 9 (30.0) | 6 (35.4) | 0.9 |
| 25–29 | 6 (20.0) | 2 (11.8) | |
| 30–34 | 12 (40.0) | 7 (41.2) | |
| 35–37 | 3(10.0) | 2 (11.8) | |
| **Parity (median, IQR), (N = 47)** | 2(1–3) | 2 (1–3) | 0.98[b] |
| 1 | 13 (43.3) | 8 (47.1) | 0.6 |
| 2 | 8(26.7) | 4 (23.5) | |
| 3 | 5 (16.7) | 1 (5.9) | |
| 4 | 3 (10.0) | 4 (23.5) | |
| 6 | 1(3.3) | 0 (0.0) | |
| **Mode of delivery, N = 47** | | | |
| Vaginal delivery | 5 (3.3) | 2 (5.9) | 0.8 |
| Assisted Vaginal delivery | 1(80.0) | 1(82.5) | |
| Emergency Caesarean section | 24 (16.7) | 14 (11.8) | |
| **Place of birth, n = 306** | | | 0.6 |
| Inborn (booked) | 12 (6.90 | 7 (5.3) | |
| Inborn (Unbooked) | 19 (10.9) | 11(8.3) | |

[a]All analysis were chi-square, except where otherwise stated.

[b.] Mann Whitney U test;

[c] Student's t-test.

cases respectively surviving beyond the 1st and 6[th] day of admission. While 75% of moderate cases survived beyond 24 days. The mortality rate by severity was 14.0 per 1000 neonate-days, 17.8per 1000 neonate-days and 86.0 per 1000 neonate-days for mild, moderate and severe cases respectively. (Log-rank p-value < 0.001).

**Table 4. Relationship between severity of HIE and neonatal and maternal factors.**

| Severity | Mild (HIE1) N = 47(%) | Moderate (HIE2) N = 190 (%) | Severe (HIE3) N = 75 (%) | P-value |
|---|---|---|---|---|
| **Age of neonate at presentation(hours) (median, IQR), n = 312** | 29(7–59) | 34.5(12–72) | 24(8–48) | 0.4 |
| 1–6 | 11 (23.4) | 38 (20.0) | 16 (21.3) | 0.6 |
| 7–24 | 11 (23.4) | 47 (24.7) | 26 (34.7) | |
| 25–72 | 23 (48.9) | 100 (52.6) | 30 (40.0) | |
| ≥72 | 2(4.3) | 5 (2.6) | 3 (4.0) | |
| **Duration of hospital stay (hour) Median, IQR, n = 312** | 240 (144–240) | 240 (168–312) | 96(24–240) | 0.0001 |
| <24 hours | 2 (4.3) | 21 (11.1) | 24 (32.0) | < 0.001 |
| 24–72 hours | 3 (6.4) | 11 (5.8) | 12 (16.0) | |
| >72 hours | 42 (89.4) | 158 (83.2) | 39 (52.0) | |
| **Gender, n = 312** | | | | |
| Female | 13 (27.7) | 68 (35.8) | 21(28.0) | 0.4 |
| Male | 34 (72.3) | 122 (64.2) | 54 (72.0) | |
| **Gestational age at delivery (weeks), n = 312** | | | | |
| Pre-term (35-<37) | 4 (8.5) | 6 (3.2) | 4 (5.3) | 0.3 |
| Term (≥37) | 43 (91.5) | 184 (96.8) | 71(94.7) | |
| **Birth weight in gram (median, IQR), n = 312** | 3200 (2800–3600) | 3000 (2700–3400) | 3000 (2500–3400) | 0.08 |
| <2500 | 5 (10.6) | 23 (12.1) | 8 (10.7) | 0.4 |
| 2,500–3999 | 38 (80.9) | 161 (84.7) | 66 (88.0) | |
| ≥ 4000 | 4 (8.5) | 6 (3.2) | 1 (1.3) | |
| **Maternal age (years), (Inborn babies, n = 47) Mean ± SD** | 29.1± 4.04 | 28.4 ± 5.6 | 26.1 ± 5.5 | 0.4 |
| <25 | 2 (22.2) | 8 (29.6) | 5 (45.5) | 0.9 |
| 25–29 | 2 (22.2) | 4 (14.8) | 2 (18.2) | |
| 30–34 | 4 (44.4) | 12 (44.4) | 3 (27.3) | |
| 35–37 | 1 (11.1) | 3 (11.1) | 1 (9.1) | |
| **Parity (median, IQR) (Inborn, n = 47)** | 2 (2–3) | 2(1–3) | 1(1–2) | 0.1 |
| 1 | 2 (22.2) | 11 (40.7) | 8 (72.7) | 0.4 |
| 2 | 4 (36.4) | 6 (22.2) | 2 (18.8) | |
| 3 | 1 (11.1) | 5 (18.5) | 0 (0.0) | |
| 4 | 2 (22.2) | 4 (14.8) | 1 (9.1) | |
| 5 | 0 (0.0) | 0 (0.0) | 0 (0.0) | |
| 6 | 0 (0.0) | 1 (3.5) | 0 (0.0) | |
| **Mode of delivery (inborn) n = 49** | | | | |
| Vaginal delivery | 1 (11.1) | 6 (22.2) | 0 (0.0) | 0.4 |
| Assisted Vaginal delivery | 0 (0.0) | 1 (3.7) | 1 (9.1) | |
| Emergency Caesarean section | 8 (88.9) | 20 (74.1) | 10 (90.9) | |
| **Place of birth, n = 306** | | | | |
| Inborn (Booked) | 5 (11.6) | 13 (6.8) | 1 (1.4) | 0.075 |
| Inborn (Unbooked) | 5 (11.6) | 14 (7.4) | 11 (14.9) | |
| Outborn | 33 (76.7) | 162 (85.7) | 62 (83.8) | |

Based on the Kaplan-Meier survival estimates on the place of birth, babies who were classified as unbooked inborn deliveries had the worst survival experience while the booked inborn had lowest deaths. Thus, the mortality rate was 6.1 per 1000 neonate-days, 110.7 per 1000 neonate-days and 25.9 per 1000 neonate-days for booked inborn, unbooked inborn and outborn babies respectively. About 50% and 75.9% of the unbooked inborn and outborn babies respectively survived to discharge while 94.7% of the booked inborn survived and were discharged

**Table 5. Association between survival of HIE babies and maternal/neonatal characteristics.**

| Severity | Dead N = 79 (%) | Survived/Discharged N = 233 (%) | P-value[a] |
|---|---|---|---|
| **Duration of hospital stay (hour) (Median, IQR)** | 24 (24–48) | 240(216–336) | <0.0001[b] |
| **<24 hours** | 44 (55.7) | 3 (1.3) | |
| **24–72 hours** | 23 (29.1) | 3 (1.3) | |
| **>72 hours** | 12 (15.2) | 227 (97.4) | |
| Mild (HE1) | 6(17.6) | 41(7.6) | <0.0001 |
| Moderate (HIE2) | 33 (67.4) | 157 (41.8) | |
| Severe (HE3) | 40 (15.0) | 35(50.6) | |
| **Age of neonate at presentation(hours) (median, IQR)** | 22(6–48) | 38 (13–72) | 0.005[b] |
| 1–6 | 21 (26.6) | 44 (18.9) | 0.034 |
| 7–24 | 28 (35.4) | 56 (24.0) | |
| 25–72 | 28 (35.4) | 125 (53.7) | |
| ≥72 | 2 (2.5) | 8 (3.4) | |
| **Gender** | | | |
| Female | 26 (33.0) | 76 (32.6) | 0.96 |
| Male | 53 (67.1) | 157 (67.4) | |
| **Gestational age at delivery** | | | |
| Preterm | 7 (8.9) | 7 (3.0) | 0.03 |
| Term | 72 (91.1) | 226 (97.0) | |
| **Weight (gram) (median, IQR)** | 3000 (2500–3500) | 3000 (2800–3450) | 0.96 |
| <2500 | 10 (12.7) | 29 (11.2) | 0.97 |
| 2,500–3999 | 64 (82.3) | 191 (85.8) | |
| ≥ 4000 | 5 (5.1) | 13(3.0) | |
| **Maternal age (years) Mean ± SD** | 26.07 ± 5.48 | 28.9 ± 5.1 | 0.08[c] |
| <25 | 7 (46.7) | 8 (25.0) | 0.4 |
| 25–29 | 3 (20.0) | 5 (15.6) | |
| 30–34 | 4 (26.7) | 15 (46.9) | |
| 35–37 | 1 (6.7) | 4 (11.8) | |
| **Parity (median, IQR), n = 47** | 1(1–2) | 2 (1–3) | 0.01[b] |
| 1 | 11(73.3) | 10 (30.3) | 0.0098 |
| 2 | 2 (13.3) | 10 (30.3) | |
| 3 | 1 (6.7) | 5 (15.2) | |
| 4 | 1 (6.7) | 6 (18.2) | |
| 5 | 0 (0.0) | 0 (0.0) | |
| 6 | 0 (0.0) | 1 (3.0) | |
| **Mode of delivery, n = 47** | | | |
| Vaginal delivery | 1 (6.7) | 6 (18.8) | 0.31 |
| Assisted Vaginal delivery | 0 (0.0) | 2 (6.3) | |
| Emergency Caesarean section | 14 (93.3) | 24 (75.0) | |
| **Place of birth** | | | |
| Inborn (booked) | 1 (1.3) | 18 (7.9) | 0.001 |
| Inborn (Unbooked) | 15 (19.2) | 15 (6.6) | |
| Outborn | 62 (79.5) | 195 (85.5) | |
| **Facilities of birth** | | | |
| Booked (LUTH) | 1 (1.3) | 18 (7.9) | < 0.0001 |
| Unbooked (LUTH) | 15 (19.2) | 15 (6.6) | |
| Public Secondary Health facility | 0 (0.0) | 13 (5.7) | |
| Private Health facility | 26 (33.3) | 104 (45.6) | |

(*Continued*)

**Table 5.** (Continued)

| Severity | Dead N = 79 (%) | Survived/Discharged N = 233 (%) | P-value[a] |
|---|---|---|---|
| Maternity Home | 7 (9.0) | 43 (18.9) | |
| Primary Health Centre | 9 (11.5) | 20 (8.8) | |
| TBA/Home | 20(57%) | 15(43%) | |
| **Year of admission** | | | |
| 2015 | 11(13.9) | 52 (22.3) | 0.5 |
| 2016 | 18 (22.8) | 50 (21.5) | |
| 2017 | 12 (15.2) | 37 (15.9) | |
| 2018 | 12 (15.2) | 34 (14.6) | |
| 2019 | 26 (33.0) | 60 (25.8) | |

[a]Except otherwise stated, logrank test.

[b] Mann Whitney U test,

[c]Student's t-test

(P-value = 0.001) (Fig 2C). There was no statistically significant difference in the survival by year of admission (Fig 2D).

## COX proportional hazard of death among babies admitted with HIE

On univariable Cox proportional regression, babies who were classified as booked inborn had about 92% less hazard of death from HIE as compared to babies classified as unbooked inborn (HR0.08, 95%CI:0.010–0.59, P-value = 0.014). Outborn babies had 64% lesser hazards of death as compared to inborn unbooked babies (HR 0.36, 95%CI: 0.20–0.63, P-value < 0.001). Furthermore, babies that were delivered preterm had about 3-fold increased hazard of death from HIE as compared to admitted term babies (HR 2.72, 95%CI: 1.25–5.90, P-value = 0.012). The hazard of death among HIE affected babies who were aged 24–72 hours at presentation were about 53% less likely as compared to babies whose age at presentation was less than 6 hours. (HR 0.47, 95%CI:0.27–0.82, P-value = 0.009). When taking age as a continuous variable, we found that for every hour increase in age at presentation, the hazard of death decreases by 1% (HR 0.99, 95%CI: 0.979–0.996, P-value = 0.004).

For every unit increase in the parity of the mothers of the inborn babies, the hazard of death among the HIE babies decreased by 49% (HR 0.51, 95%CI: 0.270–0.997, P-value = 0.049). Other statistically significant relationship after univariable regression includes gestational age at delivery. There was no statistically significant relationship between babies' gender, maternal age, mode of delivery and the hazard of death among the babies (Table 6).

After multivariable Cox proportional hazard regression and adjusting for HIE severity, gender, weight. age at presentation, type of facility and year of admission (Table 6), we found that the hazard of death among babies with severe HIE (HIE3) was about 4 times the hazard of death among infants with mild HIE (adjHR 4.23,95%CI: 1.75–10.27, P-value = 0.001). However, there was no statistically significant difference in the hazard of death among infants with moderate HIE as compared with babies with mild HIE. Furthermore, children who were unbooked inborn had about 9-fold hazard of death as compared to booked inborn children (adjHR 10.31,95%CI: 1.32–80.00, P-value = 0.03), while babies who were delivered at facilities managed by Traditional Birth Attendants or at home had about 13-fold hazard of death as compared to babies delivered by booked women in LUTH (adjHR 13.22,95%CI: 1.41–123.93,

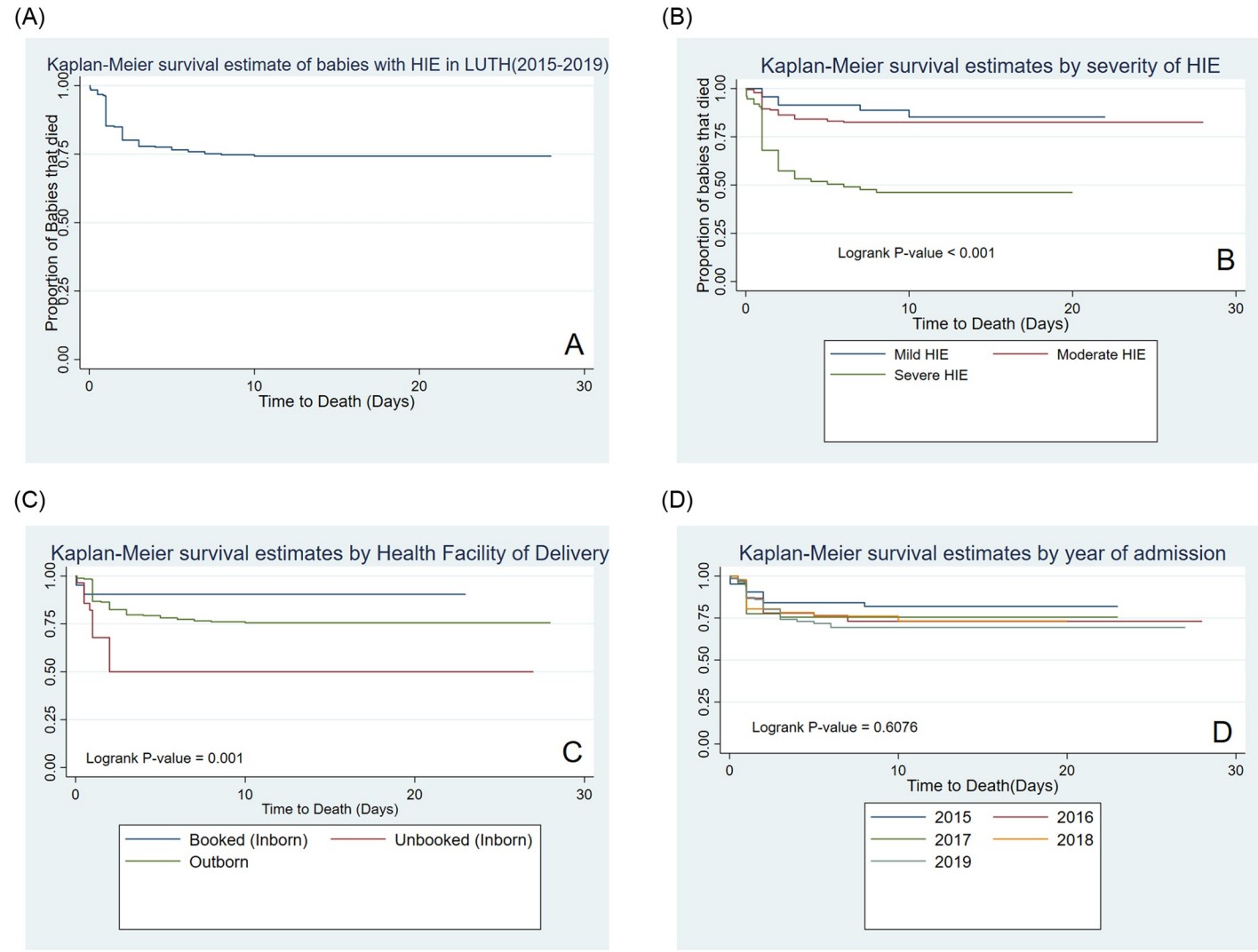

**Fig 2. The Kaplan-Meier survival experience of the cohort (A), severity of HIE (B), by place of birth (C), and year of admission (D).**

P-value = 0.024). The hazard of death increased with increasing year of admission but slightly declined in 2019.

The Schoenfeld's proportional test assumptions gave a p-value of 0.261 which showed that there was no violation of the assumptions of Cox proportional hazard. The variance inflation factor was 1.18 showing that there was no collinearity among the variables.

## Discussion

This study assessed the temporal trend in perinatal asphyxia and some of the factors associated with immediate in-hospital mortality among newborns with HIE admitted into the neonatal units of a tertiary hospital in Lagos, Nigeria, over five years. We observed an initial decline in annual asphyxia incidence from 2015 to 2017, before it increased from 2018 to 2019. Place of delivery, booking status of the mother, gestational age at birth, age of infant at presentation and severity of HIE were all found to significantly predict in-hospital mortality.

**Table 6. Univariate and multivariable regression analysis of predictors of outcome among the HIE infants.**

| Variable | Univariable | | | [a]Multivariable | | |
|---|---|---|---|---|---|---|
| | HR | 95%CI | p-value | AdjHR | 95%CI | p-value |
| **Severity of Asphyxia** | | | | | | |
| Mild (HIE1) | 1.0 | Ref | Ref | 1.0 | Ref | Ref |
| Moderate (HIE2) | 1.4 | 0.6–3.4 | 0.4 | 1.2 | 0.5–2.8 | 0.7 |
| Severe (HE3) | 5.3 | 2.2–12.4 | < 0.001 | 4.3 | 1.8–10.4 | 0.001 |
| **Age of neonate at presentation** | 0.99 | 0.979–1.0 | 0.004 | | | |
| 1–6 | 1.0 | Ref | Ref | 1.0 | Ref | Ref |
| 7–24 | 0.9 | 0.5–1.7 | 0.8 | 1.1 | 0.4–3.0 | 0.8 |
| 25–72 | 0.5 | 0.3–0.8 | 0.009 | 0.7 | 0.3–1.8 | 0.4 |
| ≥72 | 0.5 | 0.1–2.2 | 0.4 | 0.8 | 0.2–4.5 | 0.8 |
| **Gender** | | | | | | |
| Male | 1.00 | Ref | Ref | 1.0 | Ref | Ref |
| Female | 1.0 | 0.6–1.6 | 0.9 | 1.2 | 0.7–2.0 | 0.5 |
| **Weight (gram) (median, IQR)** | 0.9997 | 0.999–1.000 | 0.5 | | | |
| ≥2,500 | 1.0 | Ref | Ref | 1.0 | Ref | Ref |
| <2500 | 0.9 | 0.5–1.8 | 0.8 | 1.11 | 0.6–2.3 | 0.7 7 |
| **Facilities of birth** | . | | | | | |
| Booked (LUTH) | 1.0 | Ref | Ref | 1.0 | Ref | Ref |
| Unbooked (LUTH) | 12.9 | 1.7–98.1 | 0.01 | 10.3 | 1.3–80.01 | 0.03 |
| Public Secondary / primary Health facility | 4.1 | 0.5–32.5 | 0.18 | 4.01 | 0.4–39.1 | 0.2 |
| Private Health facility | 3.70 | 0.50–27.28 | 0.199 | 3.11 | 0.34–28.01 | 0.31 |
| Maternity Home | 2.50 | 0.31–20.31 | 0.39 | 1.92 | 0.19–19.73 | 0.58 |
| TBA/Home | 13.83 | 1.85–103.17 | 0.01 | 13.22 | 1.41–123.93 | 0.02 |
| **Year of admission** | | | | | | |
| 2015 | 1.0 | Ref | Ref | 1.0 | Ref | Ref |
| 2016 | 1.53 | 0.72–3.25 | 0.26 | 1.86 | 0.84–4.07 | 0.124 |
| 2017 | 1.47 | 0.65–3.33 | 0.36 | 2.12 | 0.89–5.08 | 0.09 |
| 2018 | 1.51 | 0.67–3.43 | 0.32 | 2.76 | 1.17–6.47 | 0.02 |
| 2019 | 1.76 | 0.87–3.57 | 0.115 | 2.65 | 1.23–5.71 | 0.01 |
| **Gestational age at delivery (weeks)** | | | | | | |
| Term (≥37) | 1.00 | Ref | Ref | | | |
| Pre-term | 2.72 | 1.25–5.9 | 0.01 | | | |
| **Place of birth** | | | | | | |
| Inborn (Unbooked) | 1.0 | Ref | Ref | | | |
| Inborn (booked) | 12.83 | 1.69–97.16 | 0.01 | | | |
| Outborn | 4.60 | 0.64–33.19 | 0.13 | | | |
| **Maternal age (years)** | 0.93 | 0.84–1.02 | 0.11 | | | |
| <25 | 1.0 | Ref | Ref | | | |
| 25–29 | 0.71 | 0.18–2.75 | 0.62 | | | |
| 30–34 | 0.39 | 0.11–1.33 | 0.13 | | | |
| 35–37 | 0.42 | 0.05–3.40 | 0.41 | | | |
| **Parity** | 0.51 | 0.26 0.98 | 0.04 | | | |
| 1 | 1.00 | Ref | Ref | | | |
| 2 | 0.25 | 0.05–1.11 | 0.07 | | | |
| ≥3 | 0.23 | 0.05–1.05 | 0.06 | | | |
| **Mode of delivery** | | | | | | |
| Vaginal delivery | 1.00 | Ref | Ref | | | |

*(Continued)*

**Table 6.** (Continued)

| Variable | Univariable | | | aMultivariable | | |
|---|---|---|---|---|---|---|
| | HR | 95%CI | p-value | AdjHR | 95%CI | p-value |
| Emergency Caesarean section | 3.72 | 0.49–28.28 | 0.21 | | | |
| **Place of birth** | | | | | | |
| Inborn (Unbooked) | 1.0 | Ref | Ref | | | |
| Inborn (booked) | 0.08 | 0.10–0.59 | 0.01 | | | |
| Outborn | 0.36 | 0.20–0.63 | < 0.001 | | | |

HR—Hazard ratio; AdjHR—Adjusted hazard ratio; CI—Confidence interval; TBA—Traditional birth attendant.

aModel adjusted for severity, gender, weight. Age at presentation, type of facility, year of admission

There is national and regional variation in the prevalence and fatality rate of HIE in literature, from less than 1% in developed countries to as high as 25% in some poor countries, possibly due to the varying quality of maternity care [3–5, 19–21]. The prevalence of HIE was 7.1% among newborn admissions in the present study with a case fatality rate of 25.3%. This is a hospital-based data and it is higher than the prevalence reported in most high income countries but similar to what poorer African nations reported in hospital-based studies too [4, 20–23]. Though the prevalence was also higher than the 3.3% reported by Ugwu et al [6] in Niger Delta, Nigeria, it is lower than the 21.1% and 24.7% reported by Ila et al [21] in Zamfara state, and Aliyu et al [24] in Kebbi state, both in Nigeria. These disparities may be related to the variation in the quality of obstetric and neonatal care within the country [5, 19–21, 24, 25]. On the other hand, the case fatality rate of 25.3% among our cohort of babies was similar to that reported by Ugwu et al (27.3%) [6] and Ila et al (25.5%) [21], but higher than that reported by Ogunkunle et al (14.7%) [26], all hospital-based studies in Nigeria. The higher case fatality rate recorded in our study may be due to our inclusion criteria. Only babies with HIE were included in our study unlike other similar studies in Nigeria that recruited infants based on Apgar scoring system [6, 11, 17–19]. The use of APGAR scores to determine asphyxia is marred by several well-documented limitations. [27, 28] In affluent societies with good facilities for impeccable neonatal care, HIE and its associated morbidities and mortalities have drastically reduced such that the case fatality rate of HIE was minimal [22, 23, 25]

Appropriate training in neonatal resuscitation can reduce neonatal mortality thereby considerably reducing the under-five mortality trends, especially in resource-poor countries such as Nigeria [29]. A study by Draiko et al [30] in Sudan highlighted the importance of healthcare provider training in preventing neonatal asphyxia. There have been several systematic reviews on the effect of newborn resuscitation training of health workers on birth asphyxia and all confirmed that training of care providers on newborn resuscitation improves and reduces neonatal morbidity and mortality [31, 32]. In Nigeria, activities around newborn resuscitation training and neonatal care increased markedly from 2010 [15, 16] In Lagos, the years 2016–2018 marked the peak of neonatal resuscitation and "helping babies breathe" training with nearly all the health facilities in Lagos (both private and publicly funded health facilities) benefiting from the series of trainings. Our study showed a reduction in the incidence per 100 admissions of asphyxiated infants in our hospital during the same periods of the neonatal resuscitation training in Lagos (2015–2017). Since more than 80% of babies admitted for asphyxia at our centre were referred from other health facilities, the observed reduction may suggest a population or community level decline in HIE for the period. There was a surge in the incidence per 100 admissions of HIE at our centre from 2018–2019. Although, an extended period of surveillance of more than 10 years will be useful to better evaluate the trends,

nonetheless, this upward trend in HIE cases seen in the last two years of this study calls for concerted efforts to arrest the trend to reduce morbidity and mortality from asphyxia. This trend will also set back all the gains achieved in reducing neonatal mortality rate in Nigeria and further derail the sustainable Development Goals (SDG) 3.2. Our result highlights the need for Nigeria to strive towards reducing child mortality through a reduction in neonatal mortality [29, 33]. Furthermore, our study suggests that training and retraining of healthcare providers on newborn resuscitation skills is necessary to reduce newborn mortality. Stakeholders should make provisions in the training guidelines of healthcare workers that requires newborn care providers to repeat neonatal resuscitation training at intervals. The neonatal resuscitation guidelines of the American Academy of Pediatrics strongly advise a two-yearly certification and refresher trainings on neonatal resuscitation skills for providers [34]. Nigeria may borrow a leaf from the American guideline, as it will ensure a workforce that is confident and adequately equipped to care for these vulnerable infants at birth.

Since the neonatal period is the most vulnerable time for a child's survival [29], our study highlighted the vulnerability of the first day of life and the high risk for mortality it imposed on asphyxiated newborns. Our study reported 14% mortality within the first 24hours of life and this was corroborated by other studies that reported similar trends [11, 26, 35]. As expected, the survival of the babies was inversely related to the HIE severity. Our study noted that the hazard of death in infants that suffered severe HIE was about 4 times the hazard of death among infants with mild asphyxia. The mortalities noted with mild HIE in this study is concerning as it is contrary to the current evidence of mortalities in mild HIE, it should be remembered that our study was retrospective in nature and that the diagnosis and classification of the HIE category was the one done at admission. It is therefore, possible that some of the infants classified as mild HIE at admission may had progressed to the severer forms of HIE but were improperly documented. Also, the accuracy of neurological assessments made at the time of admission and with respect to age of the neonate may not be correct. Diagnosing mild HIE beyond 48 hours is challenging as it could also indicate the recovery state of moderate-to-severe encephalopathy. Four out of the six infants that died in the mild HIE category presented to our facility beyond 24 hours of life. Nevertheless, it is pertinent to stress that mild forms of HIE can progress to mortality. This is important because many interventions for HIE also tend to exclude HIE stage 1 [36, 37]. For example, the international guidelines for therapeutic hypothermia for facilities with the resources for this treatment excludes infants with HIE 1. Though management of HIE in our center is still supportive, [14] in developed countries that undertake body cooling for HIE, cooling is usually instituted for HIE 2 and 3 only. The implication is that these infants with initial mild diseases miss out on treatment opportunities because they did not meet the treatment criteria within the first 6 hours of life. However, with the progression of secondary energy failure which can last up to 72 hours these mild diseases may deteriorate to severe HIE and ultimate demise. There is a need to constantly monitor these infants and possibly institute other adjuvant treatments such as Erythropoietin for them. This is even more pertinent in developing countries like Nigeria where more than 60% of births take place in the home or under the care of unskilled birth attendants [15, 16] and the asphyxiated infants present late to the hospital [33].

The present study showed that majority of the infants with HIE were outborn and that most of the babies delivered at home or in a traditional birth attendant's place, who presented with HIE, died. In the same vain peripartum referral of a high-risk pregnancy at the point of delivery is fraught with dangers. Though many of the outcome of such pregnancies come out favourable after expert management but a great majority suffer adverse events. Miyoshi in Zambia noted that cesarean sections performed emergently in referred mothers led to poorer perinatal outcomes in the infants [38].

Our study noted that the survival experience of babies of unbooked mothers in our facility is very poor with only half of the severely asphyxiated neonates from such deliveries surviving beyond two days. This is similar to other studies in Nigeria and elsewhere that documented this phenomenon [24, 38, 39] We further documented a 10-fold hazard of death among infants who were unbooked inborn as compared to booked inborn children. Conversely, inborn neonates of booked mothers had about 91% lesser hazard of death from asphyxia. This is significant and buttresses the point that high-risk pregnancies should be identified early and referred to facilities where the mother can be adequately managed and stabilized long before delivery. A triad of maternal, placental and fetal conditions in-utero can predispose to adverse neonatal outcomes expressed as neonatal encephalopathy. Placental diseases have been grouped within the great obstetrical syndromes [40, 41] such as preeclampsia, fetal growth restriction, prematurity and placental accreta spectrum, based on first-trimester abnormal placentation resulting either in chronic placental diseases (ie malperfusion syndromes) or acute intrapartum events such as abruptio placenta [42, 43]. Poor documentation of maternal history and obstetric characteristics of the pregnancies that was prevalent in the reviewed records may have prevented proper analysis of all the risk factors associated with HIE in the present study. Intrapartum HIE is in fact uncommon exclusively, and when it occurs more likely in association with antepartum more than intrpartum brain injury because of chronic placental diseases [44], given the loss of the peripheral chemoreflex [45]. Inflammatory diseases play a large role in the expression of prematurity as well as neonatal encephalopathy. Mimicry of intrapartum HIE may in fact be inflammatory responses on the maternal or fetal surfaces of the placenta causing both asphyxia and inflammatory mediator injuries before and during labor and delivery [43]. This is a major factor that suggests why therapeutic hypothermia is effective in only 1 out of 8 neonates [46]. This study did not examine nor differentiate the individual causes of HIE included in the present study, though all the infants had pertinent history of delayed cry at birth or required extensive resuscitation at birth.

Lack of medical equity and misassignment to lower levels of maternal care resulting in inadequate or inappropriate fetal surveillance can largely contribute to adverse outcomes. This is particularly experienced in low and middle income countries such as Nigeria. In Nigeria, health care financing is largely out of pocket, majority of parturients cannot afford adequate antenatal and intrapartum care. Indeed, two-third of deliveries in Nigeria occur at home or in the TBAs place with associated perinatal morbidity and mortality [15, 16] Our study found that delivery at home or in a TBA is associated with a 13-fold hazard of death as compared to inborn babies delivered by booked women among our cohort of asphyxiated babies. Home and TBAs are not equipped for providing immediate interventions to a newborn that require resuscitation at birth and they prolong the time taken to receive appropriate care thereby worsening the brain injury.

## Limitations and strengths of the study

Our study did not examine the co-morbidities of the babies while on admission. Other morbidities may contribute to the recorded mortalities. Another limitation of this study was its retrospective nature, and, as such it has all the challenges such as the inability to retrieve all demographic data for outborn mothers. There was also a possibility that some recovery phases of severe HIE were misdiagnosed as mild HIE due to late presentation and lack of specialized laboratory diagnostic support. Our study was a hospital -based study as opposed to a population-based study that is the gold standard for calculating incidence or prevalence of disease. There may also be referral bias among our cohort of subjects. However, despite these limitations, our study was the first in Nigeria to utilize a retrospective cohort study design with

appropriate statistical analysis to evaluate the hazard of death from HIE. This study was also the first to study the trends in perinatal asphyxia incidence in Nigeria and attempt to link the findings with neonatal resuscitation training while demonstrating the need for regular and recurrent training of providers of newborn care. Our study focused only on infants with clinical evidence of acute neurological dysfunction or complications of perinatal asphyxia diagnosed using standard instruments for HIE, thereby bypassing the pitfalls inherent in diagnosing perinatal asphyxia with Apgar scoring.

Overall, our study findings suggest that sustaining newborn resuscitation skills, as well as proper management of perinatal complications, are critical for saving newborn lives and reducing neonatal mortality in Nigeria. Future research should focus on analyzing the trimester-specific causes of fetal brain injuries that result in costly human and economic consequences. Prospective cohort studies can be designed to obviate some limitations of this retrospective study.

## Conclusion

Over the study period, the prevalence of HIE among the cohort of newborns admitted in our hospital was high with an initial decline and subsequent upsurge in incidence. About four fifths of the admissions with HIE were outborn babies. Their mortality rate was four-fold that of inborn babies suggesting a referral bias for mortality. Community level interventions including skilled birth attendants at delivery, newborn resuscitation trainings for healthcare personnel and capacity building for specialized care should be intensified to reduce the incidence, morbidity and mortality from HIE

## Supporting information

**S1 Table. Trends in admission, asphyxia and fatality.**
(DOCX)

**S2 Table. Life table of the survival experience of the babies with HIE.**
(DOCX)

## Acknowledgments

The LUTH neonatal unit nurses and doctors for their invaluable support and assistance throughout the data collection process.

## Author Contributions

**Conceptualization:** Beatrice Nkolika Ezenwa, Gbenga Olorunfemi, Chinyere Ezeaka.

**Data curation:** Beatrice Nkolika Ezenwa, Iretiola Fajolu, Toyin Adeniyi, Khadijah Oleolo-Ayodeji, Blessing Kene-Udemezue.

**Formal analysis:** Beatrice Nkolika Ezenwa, Gbenga Olorunfemi.

**Investigation:** Joseph A. Olamijulo.

**Methodology:** Beatrice Nkolika Ezenwa, Gbenga Olorunfemi, Iretiola Fajolu, Toyin Adeniyi, Khadijah Oleolo-Ayodeji, Joseph A. Olamijulo, Chinyere Ezeaka.

**Project administration:** Beatrice Nkolika Ezenwa, Iretiola Fajolu, Khadijah Oleolo-Ayodeji, Chinyere Ezeaka.

**Resources:** Beatrice Nkolika Ezenwa, Toyin Adeniyi, Khadijah Oleolo-Ayodeji, Blessing Kene-Udemezue, Chinyere Ezeaka.

**Software:** Gbenga Olorunfemi.

**Supervision:** Beatrice Nkolika Ezenwa, Gbenga Olorunfemi, Iretiola Fajolu, Joseph A. Olami-julo, Chinyere Ezeaka.

**Validation:** Beatrice Nkolika Ezenwa, Blessing Kene-Udemezue.

**Visualization:** Beatrice Nkolika Ezenwa, Gbenga Olorunfemi, Iretiola Fajolu, Toyin Adeniyi, Blessing Kene-Udemezue, Joseph A. Olamijulo.

**Writing – original draft:** Beatrice Nkolika Ezenwa.

**Writing – review & editing:** Beatrice Nkolika Ezenwa, Gbenga Olorunfemi, Iretiola Fajolu, Toyin Adeniyi, Khadijah Oleolo-Ayodeji, Blessing Kene-Udemezue, Joseph A. Olamijulo, Chinyere Ezeaka.

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
