## [Decision Letter · Decision Letter 0]

15 Dec 2020

PONE-D-20-22771

Trends and predictors of in-hospital mortality among babies with hypoxic ischaemic encephalopathy at a tertiary hospital in Nigeria: A retrospective cohort study

PLOS ONE

Dear Dr. Ezenwa,

Thank you for submitting your manuscript to PLOS ONE. After careful consideration, we feel that it has merit but does not fully meet PLOS ONE’s publication criteria as it currently stands. Therefore, we invite you to submit a revised version of the manuscript that addresses the points raised during the review process.

We look forward to receiving your revised manuscript.

Kind regards,

Ju Lee Oei

Academic Editor

PLOS ONE

Journal Requirements:

2.) Thank you for stating the following in the Acknowledgments Section of your manuscript:

'GO is funded by the Glaxo Smith Kline/ Sub-Saharan Africa Consortium for Advanced Biostatistics training/DELTAS Africa Fellowship through the School of Public Health, University of Witwatersrand. The views expressed are those of the authors and not necessarily that of the funders.'

'The authors received no specific funding for this work'

Reviewers' comments:

Reviewer's Responses to Questions

**Comments to the Author**

1. Is the manuscript technically sound, and do the data support the conclusions?

Reviewer #1: Partly

Reviewer #2: Yes

Reviewer #3: Yes

2. Has the statistical analysis been performed appropriately and rigorously? 

Reviewer #1: Yes

Reviewer #2: Yes

Reviewer #3: Yes

3. Have the authors made all data underlying the findings in their manuscript fully available?

Reviewer #1: No

Reviewer #2: Yes

Reviewer #3: Yes

4. Is the manuscript presented in an intelligible fashion and written in standard English?

Reviewer #1: Yes

Reviewer #2: Yes

Reviewer #3: Yes

5. Review Comments to the Author

Reviewer #1: - The research article has some implications for providing trends of neonatal encephalopathy/HIE in Nigeria.

But there is no demarcation described if the cases were NE or HIE.

- The predictors of outcome need much more elaboration- in terms of the maternal morbidities, the individual criteria used to diagnose encephalopathy, the number of neonates without adequate resuscitation, post-natal level of supportive care provided, the cause of death in mild HIE neonates etc.

- Some of the statistical analysis methods have questionable significance (specific queries are attached in the manuscript). Statistical charts/graphs could not be assessed.

- Severity of HIE described in detail as one of the predictor of fatality self-implies with more fatality and does not add much information to current literature. The mortalities in mild HIE cases need more explanation, what is authors comment on the accuracy of neurological assessments made at the time of admission and with respect to age of the neonate. Diagnosing mild HIE beyond 48 hours could also indicate the recovery state of moderate-to severe encephalopathy..

However, the research of any kind in this field in Nigeria will have implications in promoting better care of neonates with birth asphyxia.

Reviewer #2: Ezennwa et. al presented results with discussion of temporal trend analysis and retrospective cohort study over a 5-year period of asphyxiated neonates admitted to the neonatal unit of a Nigerian Hospital. Among their major findings were a median age at admission of 26.5 hours, male to female ration of 2.1:1, and 84% outworn status. Prevalence and fatality rates respectively for HIE were 7.1% and 25.3%, with fluctuating incidence over the 5 years studied.

The results presented underscore the need for worldwide concerns for mortality and morbidity as a result of hypoxic-ischemic brain injuries, particularly in developing countries. The authors correctly emphasize the importance of improved obstetrical practices and resuscitative interventions. However, they need to expand their discussion section and include for the readership the distal and proximal risk factors that impact the success of these practices and later expression of neonatal encephalopathy (NE) with neurologic sequelae (ACOG Task Force, revised 2019). The authors need to address the diagnostic analyses that integrate maternal placental and fetal conditions affecting the triad with neonatal factors expressed as NE.

Placental diseases have been grouped within the great obstetrical syndromes (DiRenzo 2009, Brosens 2011)(eg. preeclampsia, fetal growth restriction, prematurity and placental accreta spectrum) based on first-trimester abnormal placentation resulting either in chronic placental diseases (ie malperfusion syndromes) or acute intrapartum events such as abruptio placenta (Ananth 2010).

Dual horizontal and vertical diagnostic analyses allow the diagnostician to evaluate the triad as a maturing phenotype across three trimesters, expressing adaptive or maladaptive systems-biological mechanisms that preserve fetal health or promote disease (Scher 2019). Intrapartum HIE is in fact uncommon exclusively, and when it occurs more likely in association with antepartum more than intrpartum brain injury because of chronic placental diseases (Turner, 2020), given the loss of the peripheral chemoreflex (Lear 2018).

Inflammatory diseases play a large role in the expression of prematurity as well as NE. Mimicry of intrapartum HIE may in fact be inflammatory responses on the maternal or fetal surfaces of the placenta causing both asphyxia and inflammatory mediator injuries before and during labor and delivery (Scher 2020). This is a major factor that suggests why therapeutic hypothermia is effective in only 1 out of 8 neonates (McIntyre 2015).

Lack of medical equity and misassignment to lower levels of maternal care because of a lack of adequate fetal surveillance contribute to adverse outcomes. This is particularly experienced in developing countries, although health disparities exist in the most developed nations!

This expanded discussion may help with future research efforts by these authors if more MPF triad data and placental findings can be analyzed and reported from their hospital. They should be commended for their efforts given how concerning trimester-specific causes of fetal brain injuries result in costly human and economic consequences.

Reviewer #3: Hypoxic ischaemic encephalopathy (HIE) is the second cause of neonatal mortality worldwide and the first in many settings in Africa. This manuscript addresses one of the most causes of preventable deaths in neonates. Nigeria has among the largest number of neonatal deaths due to hypoxic ischaemic encephalopathy and has started a program to reduce preventable neonatal deaths.

This manuscript is a report of a retrospective study of a hospital based study. The study was conducted in Lagos University Teaching Hospital (LUTH) from 1 January 2015 to 31st December, 2019. LUTH is one of the three publicly funded tertiary health institutions in Lagos, South-Western Nigeria and has a very large NICU (80 cots).

I find two major problems. 1) The data on prevalence is confusing as this is a hospital-based study. It should be clearly stated in the Methods. Most importantly, in the Discussion, the authors must be very careful when comparing data on prevalence or incidence of HIE as some studies are population-based but others are hospital-based. Similarly, case fatality rate depends on the population selected. 2) There should be a careful selection of the references with more focus on critically addressing references of international interest from high impact journals rather than local or selected experiences/reports or non-peer-reviewed reviews.

Introduction

The Introduction is too long and not focused. Important references from higher impact journals are missing. Many of the references are non-peer-reviewed, local, or just not the best on the subject. A thorough review of the literature to address the high level of publications on the contribution of HIE to neurodevelopmental outcomes and the prevention or immediate treatment is needed.

Methods

The Methods are well-described but they should emphasize that this is a hospital-based study. It is obvious but it is important to address it as a population-based study would be ideal but not realistic. There are referral biases so the incidence is calculated per neonatal admission. The Discussion should address this better.

It is not stated how was HIE defined. The Sarnat and Sarnat exam is mentioned but this exam cannot rule out other encephalopathies.

This is a hospital-based study but referral patterns and potential changes over time are not addressed.

Was therapeutic hypothermia used?

Results

It would be important to know the unbooked rate of the inborn babies admitted for non HIE reasons to see unbooked is related to HIE.

Some results are reported with two decimal points; one is sufficient.

Table 2 which addresses inborn infants has one row of data on outborn deliveries.

Figures 1a and 1b are not necessary.

The colors and font size selection of Figure 3 make the figure difficult to read and understand the data.

Table 4 should include the N for each of the two columns on the top but if the N varies, then it should be per cell.

It is important to use terms that describe associations rather than causality. Terms such as “influence” for example in “Babies’s gender, maternal age and mode of delivery did not influence the hazard of death among the babies (Table 5)” should not be used.

Discussion

This study has many limitations which should be addressed but most importantly, interpret the results based on those limitation.

First paragraph. Similarly, “prediction” should not be used as the sample size is rather small tom perform prediction studies and the authors have not done this. The study describes associations so related terms are acceptable but prediction is not.

The discussion on prevalence is quite confusing as rates are reported by population based in some studies and by hospital admissions in others. This is like comparing apples and oranges.

Rather than provide two references of when resuscitation training and neonatal care “increased markedly” in Nigeria (around 2010, which is unrelated to this study’s time frame), the authors should reference the evidence that support each of these interventions to prevent HIE.

The Discussion needs to be focused.

Minor comments

APGAR is misspelled. The correct spelling is Apgar.

6. PLOS authors have the option to publish the peer review history of their article (what does this mean?). If published, this will include your full peer review and any attached files.

Reviewer #1: No

Reviewer #2: **Yes: **Mark S Scher

Reviewer #3: **Yes: **Waldemar A Carlo

---

## [Author Response · Author response to Decision Letter 0]

4 Feb 2021

Department of Paediatrics,

College of Medicine, 

University of Lagos

31st January, 2020

Dear Editor, 

PLoS One.

Response to reviewers’ comments on the manuscript titled ‘ Trends and predictors of in-hospital mortality among babies with hypoxic ischaemic encephalopathy at a tertiary hospital in Nigeria: a retrospective cohort study’

We are grateful for considering the above titled manuscript for possible publication in your prestigious journal. We have addressed all the comments of the reviewers as outlined below

Academic editor queries

1.Please ensure that your manuscript meets PLOS ONE's style requirements, including those for file naming. 

Authors’ response: We have thoroughly reviewed the author’s guide and ensured that the we adhered strictly to the guide

Editors’s remark: Thank you for stating the following in the Acknowledgments Section of your manuscript:

'GO is funded by the Glaxo Smith Kline/ Sub-Saharan Africa Consortium for Advanced Biostatistics training/DELTAS Africa Fellowship through the School of Public Health, University of Witwatersrand. The views expressed are those of the authors and not necessarily that of the funders.'

'The authors received no specific funding for this work'

.

Authors’ response: Although Dr Gbenga Olorunfemi (GO) is currently funded by the Glaxo Smith Kline/ Sub-Saharan Africa Consortium for Advanced Biostatistics training/DELTAS Africa Fellowship through the School of Public Health, University of Witwatersrand, We did not receive specific funding for this project as the study was self-funded. Since this acknowledgement is against the rules of PLOS one, we have removed it.

Reviewer 1

Reviewer 1 Comment: The research article has some implications for providing trends of neonatal encephalopathy/HIE in Nigeria. But there is no demarcation described if the cases were NE or HIE

Authors’ response: We thank you for your encouraging comment. With respect to distinguishing between NE or HIE, we were not able to differentiate neonatal encephalopathy from HIE in this study due to: 

1. This was a retrospective study; we can’t change the diagnosis. 

2. Predictive imaging studies such as MRI and laboratories markers of severe asphyxia as well as arterial blood gas analysis are not yet routinely available in our hospital. 

It is however pertinent to mention that all the infants had antecedent history of delayed cry at birth or required extensive resuscitation at birth.

Reviewer 1 Comment: The predictors of outcome need much more elaboration- in terms of the maternal morbidities, the individual criteria used to diagnose encephalopathy, the number of neonates without adequate resuscitation, post-natal level of supportive care provided, the cause of death in mild HIE neonates etc.

Authors’ response: Being a retrospective study with majority (80%) of the babies being outborn, it was difficult to obtain accurate morbidity history of the mothers beyond what we analyzed. In our centre, majority of parents and guardians do not consent to autopsy to confirm cause of death. Thus, such factor may not be validly analyzed. 

Reviewer 1 Comment: Some of the statistical analysis methods have questionable significance (specific queries are attached in the manuscript). Statistical charts/graphs could not be assessed.

Authors’ response: Specific queries have been addressed in the manuscript. Statistical charts/graphs were separately uploaded according to instructions of the journal

Reviewer 1 Comment: Severity of HIE described in detail as one of the predictor of fatality self-implies with more fatality and does not add much information to current literature. 

Authors’ response: We agree that severity of HIE is a known predictor of fatality. However, our study contributes to knowledge by highlighting the global magnitude of variability in hazard of death related to HIE severity. 

Reviewer 1 Comment: The mortalities in mild HIE cases need more explanation, what is authors comment on the accuracy of neurological assessments made at the time of admission and with respect to age of the neonate. Diagnosing mild HIE beyond 48 hours could also indicate the recovery state of moderate-to severe encephalopathy.

Authors’ response: The mortalities noted with mild HIE in this study is concerning as it is contrary to the current evidence of mortalities in mild HIE, it should be remembered that our study was retrospective in nature and that the diagnosis and classification of the HIE category was the one done at admission. It is therefore, possible that some of the infants classified as mild HIE at admission may had progressed to the severer forms of HIE but were improperly documented. Also, the accuracy of neurological assessments made at the time of admission and with respect to age of the neonate may not be correct. As the reviewer rightly noted, diagnosing mild HIE beyond 48 hours is challenging as it could also indicate the recovery state of moderate-to severe encephalopathy. Four out of the six infants that died in the mild HIE category presented to our facility beyond 24 hours of life.

Reviewer 1 Comment: However, the research of any kind in this field in Nigeria will have implications in promoting better care of neonates with birth asphyxia.

Authors’ response: Thank you for this remark. Indeed, we are optimistic that the data and robust analysis presented will contribute to knowledge and management policy in Nigeria and most lower and middle income countries with similar demography and health system like Nigeria. Furthermore, future meta-analysis and reviews will be able to include published data such as our study from Nigeria 

Reviewer 2 

Reviewer 2. Ezennwa et. al presented results with discussion of temporal trend analysis and retrospective cohort study over a 5-year period of asphyxiated neonates admitted to the neonatal unit of a Nigerian Hospital. Among their major findings were a median age at admission of 26.5 hours, male to female ration of 2.1:1, and 84% outworn status. Prevalence and fatality rates respectively for HIE were 7.1% and 25.3%, with fluctuating incidence over the 5 years studied.

The results presented underscore the need for worldwide concerns for mortality and morbidity as a result of hypoxic-ischemic brain injuries, particularly in developing countries. The authors correctly emphasize the importance of improved obstetrical practices and resuscitative interventions. However, they need to expand their discussion section and include for the readership the distal and proximal risk factors that impact the success of these practices and later expression of neonatal encephalopathy (NE) with neurologic sequelae (ACOG Task Force, revised 2019). The authors need to address the diagnostic analyses that integrate maternal placental and fetal conditions affecting the triad with neonatal factors expressed as NE

Authors’ response: We thank the reviewer for the remark. We have now discussed the points mentioned in the discussion section of the manuscript. (See lines 420- 449.)

Reviewer 2 Comment: Placental diseases have been grouped within the great obstetrical syndromes (DiRenzo 2009, Brosens 2011) (eg. preeclampsia, fetal growth restriction, prematurity and placental accreta spectrum) based on first-trimester abnormal placentation resulting either in chronic placental diseases (ie malperfusion syndromes) or acute intrapartum events such as abruptio placenta (Ananth 2010). Dual horizontal and vertical diagnostic analyses allow the diagnostician to evaluate the triad as a maturing phenotype across three trimesters, expressing adaptive or maladaptive systems-biological mechanisms that preserve fetal health or promote disease (Scher 2019). Intrapartum HIE is in fact uncommon exclusively, and when it occurs more likely in association with antepartum more than intrpartum brain injury because of chronic placental diseases (Turner, 2020), given the loss of the peripheral chemoreflex (Lear 2018).

Inflammatory diseases play a large role in the expression of prematurity as well as NE. Mimicry of intrapartum HIE may in fact be inflammatory responses on the maternal or fetal surfaces of the placenta causing both asphyxia and inflammatory mediator injuries before and during labor and delivery (Scher 2020). This is a major factor that suggests why therapeutic hypothermia is effective in only 1 out of 8 neonates (McIntyre 2015). Lack of medical equity and misassignment to lower levels of maternal care because of a lack of adequate fetal surveillance contribute to adverse outcomes. This is particularly experienced in developing countries, although health disparities exist in the most developed nations. 

This expanded discussion may help with future research efforts by these authors if more MPF triad data and placental findings can be analyzed and reported from their hospital. They should be commended for their efforts given how concerning trimester-specific causes of fetal brain injuries result in costly human and economic consequences.

Authors’ response: We thank the reviewer for the remarks. We have now discussed some of the points mentioned in the discussion section of the manuscript. (See lines 424-449). Furthermore, future prospective projects will be designed to incorporate the comments and idea. Thank you 

Reviewer 3

Reviewer 3 Comment: Hypoxic ischaemic encephalopathy (HIE) is the second cause of neonatal mortality worldwide and the first in many settings in Africa. This manuscript addresses one of the most causes of preventable deaths in neonates. Nigeria has among the largest number of neonatal deaths due to hypoxic ischaemic encephalopathy and has started a program to reduce preventable neonatal deaths.

This manuscript is a report of a retrospective study of a hospital-based study. The study was conducted in Lagos University Teaching Hospital (LUTH) from 1 January 2015 to 31st December, 2019. LUTH is one of the three publicly funded tertiary health institutions in Lagos, South-Western Nigeria and has a very large NICU (80 cots).

Authors’ response: We thank you for your remark

Reviewer 3 Comment: I find two major problems. 1) The data on prevalence is confusing as this is a hospital-based study. It should be clearly stated in the Methods. Most importantly, in the Discussion, the authors must be very careful when comparing data on prevalence or incidence of HIE as some studies are population-based but others are hospital-based. Similarly, case fatality rate depends on the population selected. 

Authors’ response: We have inserted ‘hospital- based’ in the methods section. (see line 94). We have also reviewed our discussion section to ensure that we compared our results with similarly conducted institutional based studies. In areas where we quoted prevalence from population- based studies, we have distinguished it in the manuscript. However, we believe prevalence and incidence of a condition can also be obtained from hospital-based studies based on the definition of prevalence and incidence. 

Reviewer 3 Comment: There should be a careful selection of the references with more focus on critically addressing references of international interest from high impact journals rather than local or selected experiences/reports or non-peer-reviewed reviews

Authors’ response: In line with the suggestion of the reviewer, we have conducted further literature search to include relevant international and national articles in high impact journals. This has been reflected in the introduction and discussion. Such additional journal articles includes: references 1, 4, 31-33, 41-47.

However, we thoroughly reviewed the manuscript and found that most of the articles cited were suitable and relevant as we believe that local articles are also necessary as they discuss comparable local data. Many of the literature found in high impact journals are from studies in high income countries. We believe citing some works done locally helped us to properly contextualize the study findings. All cited articles in this study can be accessed online.

Reviewer 3 Comment: Introduction: The Introduction is too long and not focused. Important references from higher impact journals are missing. Many of the references are non-peer-reviewed, local, or just not the best on the subject. A thorough review of the literature to address the high level of publications on the contribution of HIE to neurodevelopmental outcomes and the prevention or immediate treatment is needed

Authors’ response: We have reviewed the introduction section and made further edit to reduce its length. We also conducted further literature search to improve the content of the manuscript. See further response in the penultimate authors’ response.

Reviewer 3 Comment: Methods: The Methods are well-described but they should emphasize that this is a hospital-based study. It is obvious but it is important to address it as a population-based study would be ideal but not realistic. There are referral biases so the incidence is calculated per neonatal admission. The Discussion should address this better.

Authors’ response: We have inserted “hospital-based” in the method section (see line 94). Though, population based studies are more appropriate for calculating prevalence/incidence of a disease. However, in the absence of data from population-based studies, data from hospital-based studies can show the tip of the iceberg. We agree that there could be referral bias. We have included this as a limitation of the study. See line 455. 

Reviewer 3 Comment: It is not stated how was HIE defined. The Sarnat and Sarnat exam is mentioned but this exam cannot rule out other encephalopathies.

Authors’ response: 

HIE was defined for the purposes of this study as the presence of encephalopathy or altered consciousness and multi-organ failure in a term newborn with a positive history of delayed cry at birth or required prolonged resuscitation at birth in addition to the presence of any of the neurological features as contained in the Sarnat and Sarnat classification. (See lines 112-116)

Reviewer 3 Comment: This is a hospital-based study but referral patterns and potential changes over time are not addressed.

Authors’ response: We agree that there may be changes in incidence based on changes in referral pattern. Indeed, such a perceived change in pattern is a good reason/justification for this study. A change in the incidence trends may therefore be a product of changes in referral pattern which may be a reflection of interventions at the peripheral centres. Thus, the trends in HIE from our hospital-based study can assist to know the current burden at our centre. We have added the statement below in the introduction section: “The trainings empowered the health care providers in the peripheral centers to promptly identify and refer cases of HIE that will require tertiary level care A hospital-based study on the burden of HIE can assist in prioritizing personnel and equipment to effectively manage the HIE cases that may present to the hospital. (Lines 80-81; 84-86)

Reviewer 3 Comment: Was therapeutic hypothermia used?

Authors’ response: No therapeutic hypothermia was not used as it was not available in the country currently.

Reviewer 3 Comment: Results: It would be important to know the unbooked rate of the inborn babies admitted for non HIE reasons to see unbooked is related to HIE. Some results are reported with two decimal points; one is sufficient

Authors’ response: This study was a retrospective cohort study of babies admitted for HIE over the study period. The relationship between inborn booked, inborn unbooked and outborn and severity of HIE was analysed in Table 3. Since this study was not a case control study, we did not assess nor analyze other non-HIE morbidities admitted during the study period.

We have corrected the decimal places to one in the results section

Reviewer 3 Comment: Table 2 which addresses inborn infants has one row of data on outborn deliveries

Authors’ response: Thank you for the observation. We have removed the outborn deliveries and the percentages re-calculated.

Reviewer 3 Comment: Figures 1a and 1b are not necessary.

Authors’ response: Since our main study objective was around severity and outcomes of HIE we decided to depict them in the figures. We suggest that the figures 1a and 1b be left in the manuscript

Reviewer 3 Comment: The colors and font size selection of Figure 3 make the figure difficult to read and understand the data.

Authors’ response: The graphs in figure 3 has been expanded and enlarged to make the figure clearer to understand. Thank you 

Reviewer 3 Comment: Table 4 should include the N for each of the two columns on the top but if the N varies, then it should be per cell.

Authors’ response: The N for each column of Table 4 has now been added at the top (N= 79 , N= 233)

Reviewer 3 Comment: It is important to use terms that describe associations rather than causality. Terms such as “influence” for example in “Babies’s gender, maternal age and mode of delivery did not influence the hazard of death among the babies (Table 5)” should not be used

Authors’ response: The statement has been edited to read “There was no statistically significant relationship between babies’s gender, maternal age, mode of delivery and the hazard of death among the babies (Table 5)” line 302 – line 303

Reviewer 3 Comment: Discussion: This study has many limitations which should be addressed but most importantly, interpret the results based on those limitations.

Authors’ response: In line with the comment of the reviewer, we have reviewed the discussion and further reviewed the limitations of the study. See line 457 – 465.

Reviewer 3 Comment: First paragraph. Similarly, “prediction” should not be used as the sample size is rather small tom perform prediction studies and the authors have not done this. The study describes associations so related terms are acceptable but prediction is not.

Authors’ response: In line with the comment of the reviewer, “predictor” has been changed to “the factors associated with”. Line 325-326

Reviewer 3 Comment: The discussion on prevalence is quite confusing as rates are reported by population based in some studies and by hospital admissions in others. This is like comparing apples and oranges

Authors’ response: All the cited references on prevalence and incidence are from hospital-based studies.

Reviewer 3 Comment: Rather than provide two references of when resuscitation training and neonatal care “increased markedly” in Nigeria (around 2010, which is unrelated to this study’s time frame), the authors should reference the evidence that support each of these interventions to prevent HIE

Authors’ response: Two references that supported the benefit of neonatal resuscitation training in Nigeria have been included. There was a national effort to train health care personnel at peripheral level on neonatal resuscitation from the year 2010. Such an initiative of training personnel at peripheral centres was also embraced in Lagos state from 2010. Thus, it is expected that such previous trainings, can cause a reduction in incidence of HIE in later years. This study commenced in 2015 – 2019. We noticed a reduced trend from 2015 – 2017 and thus we postulated that these trainings might have led to the reduction. Our study did not show if the reduction in HIE incidence occurred earlier than 2015 because our study commenced in 2015. 

Reviewer 3 Comment: The Discussion needs to be focused

Authors’ response: We thank the reviewer for this comment. We have reviewed the discussion again to remove redundant words.

Reviewer 3 Comment: Minor comments. APGAR is misspelled. The correct spelling is Apgar.

Authors’ response: APGAR has been changed to Apgar throughout the manuscript

REVIEWER’S COMMENTS IN THE MANUSCRIPT

 Reviewer 1 Comment in the manuscript: Could you mention on the level of care for HIE neonates at this hospital? Mechanical ventilation/ CPAP/ equipments/ supportive care/ nurse patient ratio. Neuroimaging ?

Authors’ response: The statesment below has been added to lines 111 – 115. The neonatal unit had no functional mechanical ventilator but both conventional and improvised bubble Continuous Positive Airway Pressure devices were available for infants requiring respiratory support. The hospital laboratories and imaging units also serve the neonatal wards though certain specialized investigations such as arterial blood gases and magnetic resonance imaging studies were not routinely done for patients

Reviewer 1 Comment in the manuscript: Can you give a data on the proportion of high-risk HIE births attended by neonatologist? Are the neonatologists and paediatricians different in your hospital? This can be a useful predictor of outcome after neonatal resuscitation in HIE.

Authors’ response: No the neonatologists and paediatrician are the same. All the consultants in the neonatology unit are neonatologists in addition to being Paediatricians. All high-risk deliveries were attended by Paediatricians who may be senior Registrar or the consultant in the neonatal unit. 

Reviewer 1 Comment in the manuscript: Were there any exclusion criteria based on other congenital malforamtions/maternal drugs/ other comorbidities/infections etc which would affect in-hospital mortalities?

Authors’ response : Congenital malforamtions/maternal drugs/ other comorbidities/infections were excluded from the study. This is stated in the text line 116-117

Reviewer 1 Comment in the manuscript: Please mention the staging of HIE was done by whom? 

Neonatologist/paediatrician/trainees/nurses etc

Authors’ response: As per the protocol of the unit, on the admission of each baby, the severity of HIE was determined by the senior registrar, consultant paediatrician or the neonatologist that first examined the baby based on the Sarnat and Sarnat staging. This has been inserted in Lines 126-129

Reviewer 1 Comment in the manuscript: How did you differentiate HIE from neonatal encephalopathy? Blood gas on admission/markers of asphyxia/APGARS?

Authors’ response: We were not able to conclusively differentiate neonatal encephalopathy from HIE in this study due to 1. This is a retrospective study, we can’t change the diagnosis. 2. There were challenges with availability of predictive imaging studies and laboratories markers of severe asphyxia. It is however pertinent to state that all the infants had antecedent history of delayed cry at birth or required extensive resuscitation at birth and all fulfilled the Sarnat and Sarnat criteria for HIE

Reviewer 1 Comment in the manuscript: Could you give data on admissions in first six hours of life? What is your implication of calculating admission in first 24 hours? 

Authors’ response: In line with the reviewer’s comment we have now added data on the first 6 hours of life. All the inborn neonates were admitted within 6 hours of birth but only 11 outborn infants with HIE were admitted within 6 hours of life. (Table 1and Lines 174-175)

The implication is that treatment such as therapeutic hypothermia that requires commencing within 6 hours of life may exclude majority of outborn infants who most need the treatment. With a documented evidendence that majority of the infants with HIE present late to treatment centers, more efforts will be geared towards developing and providing alternative treatments such as MgSO4, Erythropoietin etc which may be beneficial even when commenced after the 6 hour window

Reviewer 1 Comment in the manuscript : Could not assess the chart

Authors’ response: In line with the journal’s instruction, the chart and figures were uploaded separately. However, for the purpose of the review process, the figures and charts are again uploaded.

Reviewer 1 Comment in the manuscript: Could you subgroup maternal and neonatal characteristics into 2015-2017 and 2018-2019? To find the predictors of increase in HIE rates?

Authors’ response: This sub-analysis has been done in the new Table 3

Reviewer 1 Comment in the manuscript: What was the minimum gestation here?

Authors’ response: The Minimum gestation was 35 weeks. This has been shown in the Table 4

Reviewer 1 Comment in the manuscript: Please verify total number of mothers, known and unknown data in Table 3

Authors’ response: The total number of mothers (n=49) have been stated in Table 4. There were two missing values on age. The missing values have been shown all through the Tables. There were 49 mothers that delivered within the hospital (inborn) and these were the ones whose biodata were relatively complete. The biodata of the outborn mothers were very deficient hence they were not analysed

Reviewer 1 Comment in the manuscript: Data does not match with total number of HIE births

Authors’ response: The retrospective nature of the study hampered complete data collection as most of the maternal information were not found in the case files of the outborn infants and those that contain some records on maternal data were grossly deficient with missing data. On account of this only the available maternal data of inborn neonates were analysed

Reviewer 1 Comment in the manuscript: Is there any significance of calculating median parity of ‘two’ here? While the maximum number of mothers were primiparous in all category

Authors’ response: There was no significance. We rechecked our calculation, and it is still the same figures. The maximum number of parity for mothers of babies with mild HIE was para2 

Reviewer 1 Comment in the manuscript: What are the figures in brackets (88.9) here? (under mode of delivery)

Authors’ response: The figures in bracket are percentages. This has now been depicted at the top of the Table

Reviewer 1 Comment in the manuscript: What were the reasons for mortality here especially for mortality in mild HIE neonates? As this is in contrary to the current evidence of mortalities in mild HIE cases? please describe in detail in the discussion.

Authors’ response: The mortalities noted with mild HIE in this study is concerning as it is contrary to the current evidence of mortalities in mild HIE, it should be remembered that our study was retrospective in nature and that the diagnosis and classification of the HIE category was the one done at admission. It is therefore, possible that some of the infants classified as mild HIE at admission may had progressed to the severer forms of HIE but were improperly documented. Also, the accuracy of neurological assessments made at the time of admission and with respect to age of the neonate may not be correct. Diagnosing mild HIE beyond 48 hours is challenging as it could also indicate the recovery state of moderate-to severe encephalopathy. Four out of the six infants that died in the mild HIE category presented to our facility beyond 24 hours of life.

Reviewer 1 Comment in the manuscript: Was there any significant association found between mode of delivery and incidence/ fatalities in HIE in this study? 

Authors’ response: The only association can be attributed to the high risk nature of the deliveries seen in unbooked mothers which may have accounted for the increased caesarean section documented in the inborn deliveries

Reviewer 1 Comment in the manuscript: Were there any fallacies found with respect to lack of adequate resuscitation particularly in this study? 

Authors’ response: There were no fallacies identified, however, we noted that babies delivered in TBAs and at home had 14-fold increased risk of HIE as their delivery were not attended by skilled birth attendants nor anyone versed in neonatal resuscitation techniques.

---

## [Decision Letter · Decision Letter 1]

3 Mar 2021

PONE-D-20-22771R1

Trends and predictors of in-hospital mortality among babies with hypoxic ischaemic encephalopathy at a tertiary hospital in Nigeria: A retrospective cohort study

PLOS ONE

Dear Dr. Ezenwa,

Thank you for submitting your manuscript to PLOS ONE. After careful consideration, we feel that it has merit but does not fully meet PLOS ONE’s publication criteria as it currently stands. Therefore, we invite you to submit a revised version of the manuscript that addresses the points raised during the review process.

Please address the comments of both reviewers, particular those from reviewer #3. 

We look forward to receiving your revised manuscript.

Kind regards,

Ju Lee Oei

Academic Editor

PLOS ONE

Reviewers' comments:

Reviewer's Responses to Questions

**Comments to the Author**

1. If the authors have adequately addressed your comments raised in a previous round of review and you feel that this manuscript is now acceptable for publication, you may indicate that here to bypass the “Comments to the Author” section, enter your conflict of interest statement in the “Confidential to Editor” section, and submit your "Accept" recommendation.

Reviewer #2: All comments have been addressed

Reviewer #3: (No Response)

2. Is the manuscript technically sound, and do the data support the conclusions?

Reviewer #2: Yes

Reviewer #3: No

3. Has the statistical analysis been performed appropriately and rigorously? 

Reviewer #2: Yes

Reviewer #3: Yes

4. Have the authors made all data underlying the findings in their manuscript fully available?

Reviewer #2: Yes

Reviewer #3: Yes

5. Is the manuscript presented in an intelligible fashion and written in standard English?

Reviewer #2: Yes

Reviewer #3: Yes

6. Review Comments to the Author

Reviewer #2: As stated for my initial review, this is an important hospital-based study that attempts to address important and complex factors influencing neonatal morbidity (and maternal mortalities!), still more profoundly expressed in resource-poor nations! Thank-you to the authors for incorporating Reviewer 2's suggestions. Other than proofing the manuscript for occasional word corrections in tense, etc in several instances (e.g. line 48,) , this revised manuscript is acceptable for publication.

The challenges for my colleagues who present their research findings regarding maternal- child care before and after birth need constant scrutiny and attention through peer-reviewed dialogue to achieve world-wide medical equity. These challenges existed during the latter part of the 20th century following WWII and were addressed by nations with adequate research and public health resources to reduce mortalities and improve morbidities in their nations (e.g. Collaborative Perinatal Project in the US). Despite obvious successes based on these birth cohort efforts, new challenges have either replaced or sustained the former ones given lack of resources even within these nations. Partnerships among healthcare, social-support agencies and government are needed to merge priorities that will ultimately improve both medical and socioeconomic health for an entire nation's citizenry.

Good luck for future contributions.

Reviewer #3: General comments

This is a hospital-based study largely of a subpopulation of referred infants with HIE in a large city where there are other hospitals receiving referred infants. The observed trends and changes over time could be due to referral bias. This is now addressed well in the Discussion but not in the Abstract (especially in the Conclusions).

Abstract

The abstract needs to address the potential for referral bias.

This is a hospital-based study conducted retrospectively and covering many years. The authors must be more careful when addressing the incidence and fatality rate of asphyxia and refer to hospital admissions rather than make general unqualified statements such as “The annual incidence declined by 1.4% per annum while the annual fatality rate increased by 10.3% per annum from 2015 to 2019.”

About four fifths of the admissions with HIE were outborn babies. Their mortality rate was four-fold that of inborn babies. The referral bias for mortality if also extremely important.

Introduction

The Introduction is based on rather low level of evidence studies and reviews rather than strong data on HIE from well designed RCTs, cohort studies, and meta-analyses. This was commented in the previous review but not addressed. There are major studies from multi-country (LMICs) populations as well as some single country (LMIC) that should addressed as well as the meta-analysis rather than report small retrospective cohorts.

Methods

There are referral biases so the incidence is calculated per neonatal admission. The Methods and Discussion must address this well. Were there changes in referral patterns? Did the proportion of admitted babies compared to other 2 NICUs in Lagos differ? I think the best solution is to acknowledge in both the Methods and Discussion (including the Conclusions) that referral bias was possible. If it can be quantified, it would be best, but it is understood that this may be hard to quantify.

The statistical analysis must be carefully performed as referral bias can have a big impact.

Results

It would be best to compare inborn and outborn infants in Tables 1 and 2.

Figures 1a and 1b are not needed. Fig 1a does not have the n/N data that should be provided better in the text. A three d pie does not help.

Discussion

This study has many limitations which should be addressed but most importantly, interpret the results based on those limitations. The Conclusions miss the point totally.

Minor

HIE and asphyxia (e.g. Figures 2a and 2c) are used. It is best to stick to one only.

The title of Figure 2c has a typo (severtly).

7. PLOS authors have the option to publish the peer review history of their article (what does this mean?). If published, this will include your full peer review and any attached files.

Reviewer #2: **Yes: **Mark Steven Scher MD

Reviewer #3: No

---

## [Author Response · Author response to Decision Letter 1]

26 Mar 2021

Department of Paediatrics,

College of Medicine, 

University of Lagos

21st March, 2021

Dear Editor, 

PLoS One.

Dear Sir,

Response to reviewers’ comments on the manuscript titled ‘Trends and predictors of in-hospital mortality among babies with hypoxic ischaemic encephalopathy at a tertiary hospital in Nigeria: a retrospective cohort study’

Once more, we are grateful for considering the above titled manuscript for possible publication in your prestigious journal. We have addressed all the further comments of the reviewers as outlined below:

Reviewers’ comments

Reviewer #2: 

As stated for my initial review, this is an important hospital-based study that attempts to address important and complex factors influencing neonatal morbidity (and maternal mortalities!), still more profoundly expressed in resource-poor nations! Thank-you to the authors for incorporating Reviewer 2's suggestions. Other than proofing the manuscript for occasional word corrections in tense, etc in several instances (e.g. line 48,) , this revised manuscript is acceptable for publication.

The challenges for my colleagues who present their research findings regarding maternal- child care before and after birth need constant scrutiny and attention through peer-reviewed dialogue to achieve world-wide medical equity. These challenges existed during the latter part of the 20th century following WWII and were addressed by nations with adequate research and public health resources to reduce mortalities and improve morbidities in their nations (e.g. Collaborative Perinatal Project in the US). Despite obvious successes based on these birth cohort efforts, new challenges have either replaced or sustained the former ones given lack of resources even within these nations. Partnerships among healthcare, social-support agencies and government are needed to merge priorities that will ultimately improve both medical and socioeconomic health for an entire nation's citizenry.

Good luck for future contributions.

Authors’ response: Thank you for your comments, insights and encouragement

Reviewer #3: 

Reviewer #3: General comments

This is a hospital-based study largely of a subpopulation of referred infants with HIE in a large city where there are other hospitals receiving referred infants. The observed trends and changes over time could be due to referral bias. This is now addressed well in the Discussion but not in the Abstract (especially in the Conclusions).

Authors’ response: Thank you for your comments. The conclusion of the abstract has been revised to include the fact that majority of our HIE babies were outborn which may in turn impact on the pattern of the burden of the HIE at our centre. Thus, the conclusion of the abstract now states “. The case fatality rate of HIE is high and increasing at our centre and mainly driven by the pattern of admission of HIE cases among outborn babies.  Thus, community level interventions including skilled birth attendants at delivery, newborn resuscitation trainings for healthcare personnel and capacity building for specialized care should be intensified to reduce the burden of HIE from perinatal asphyxia”.

Reviewer #3 comment: Abstract

The abstract needs to address the potential for referral bias.

This is a hospital-based study conducted retrospectively and covering many years. The authors must be more careful when addressing the incidence and fatality rate of asphyxia and refer to hospital admissions rather than make general unqualified statements such as “The annual incidence declined by 1.4% per annum while the annual fatality rate increased by 10.3% per annum from 2015 to 2019.”

Authors’ response: The abstract has been revised and changes were made in line with the suggestion of the reviewer. The objective of the study was made more concise to reflect the data was collected from only one hospital.

In the results section we added phrases such as “ at our centre”, “among the hospital admissions” to reflect that our results and conclusions were based on data obtained from  our  facility.

Reviewer #3 comment: About four fifths of the admissions with HIE were outborn babies. Their mortality rate was four-fold that of inborn babies. The referral bias for mortality if also extremely important.

Authors’ response: Thank you for the kind suggestion. We have incorporated this idea into the conclusion section.

Reviewer #3 comment: Introduction

The Introduction is based on rather low level of evidence studies and reviews rather than strong data on HIE from well designed RCTs, cohort studies, and meta-analyses. This was commented in the previous review but not addressed. 

There are major studies from multi-country (LMICs) populations as well as some single country (LMIC) that should addressed as well as the meta-analysis rather than report small retrospective cohorts.

Authors’ response: The authors have once again done an extensive literature search using the search terms: HIE or hypoxic ischaemic encephalopathy or perinatal asphyxia and newborn and LMIC or low and middle income countries and RCT or systematic reviews

Majority of the articles found were not addressing our focus. Our research focused on the trend in incidence and predictors of mortality from HIE in our center, which is in a lower middle income country. Majority of the RCT and systematic reviews were looking at the treatment and interventions given for HIE making it difficult to cite many of them in our study. (Since that was not our focus). We have further reviewed and cited 3 more literatures (one from Uganda [reference 5], Brazil [reference 9] and United Kingdom [reference 7])

We would be happy to review the suitability of any literature specifically pointed out or listed by the reviewer. 

Reviewer #3 comment: Methods

There are referral biases so the incidence is calculated per neonatal admission. The Methods and Discussion must address this well. Were there changes in referral patterns? Did the proportion of admitted babies compared to other 2 NICUs in Lagos differ? I think the best solution is to acknowledge in both the Methods and Discussion (including the Conclusions) that referral bias was possible. If it can be quantified, it would be best, but it is understood that this may be hard to quantify.

Authors’ response: Our study has utilized “per admission” as the denominator of the rates in our study. Our data reports the pattern from our centre. We agree that the rates may be affected by the referral pattern. We do not have data on the admission pattern from the other 2 NICUs in Lagos. We have included this as a limitation of the study and stated it as: “Our study was a hospital -based study as opposed to a population-based study that is the gold standard for calculating incidence or prevalence of disease. There may also be referral bias among our cohort of subjects”- see line 454-456 under limitation.

Reviewer #3 comment: The statistical analysis must be carefully performed as referral bias can have a big impact.

Authors response: Although there are limitations with retrospective data, however, we have utilized robust statistical methods in this study. For the trends analysis, we reported rates per admission at our hospital and we went on to conduct multivariable Cox proportional analysis to correct for confounding variables.  

Reviewer #3 comment: Results

It would be best to compare inborn and outborn infants in Tables 1 and 2 

Authors’ response: The focus of the paper is to evaluate the predictors of mortality. “Outborn vs inborn” is one of the explanatory variables for the predictors of mortality. Outborn vs inborn is not an outcome variable in this study. Thus, the association of outborn / inborn and the major outcome was assessed in all the relevant Tables. Comparing inborn vs outborn in Tables 1 and 2 in this manuscript as suggested will change the focus of the paper as our outcome is mortality and morbidity (severity) of HIE among our cohort. Whereas, inborn/outborn is one of the explanatory variables that was assessed. The variable was also part of the multivariable model. For this manuscript to remain focused and not be unwieldy, we suggest we maintain the analysis and Tables. 

Reviewer #3 comment: Figures 1a and 1b are not needed. Fig 1a does not have the n/N data that should be provided better in the text. A three d pie does not help.

Authors’ response:  Although, we believe figures 1a and 1b will visually depict the pattern of the major outcome for the audience to appreciate the burden, and our texts complimented the tables however, in line with the reviewer’s comment, Fig 1a and 1b have been expunged.

Reviewer #3 comment: Discussion

This study has many limitations which should be addressed but most importantly, interpret the results based on those limitations. The Conclusions miss the point totally.

Authors’ response: The limitations of this study has been extensively discussed and the fact that this study was based on single hospital data has been included in all sections of the manuscript. (See line 448 “Limitation and strength of the study”) The concluding statements have been revised and modified to capture these concerns.

Reviewer #3 comment: Minor

HIE and asphyxia (e.g. Figures 2a and 2c) are used. It is best to stick to one only.

The title of Figure 2c has a typo (severtly).

Authors’ response: Thank you for these observations. ‘Asphyxia’ has been changed to HIE and the typo corrected.

Once again, we are grateful to the reviewers and editors for thorough review of the manuscript. The comments have greatly improved the manuscript. 

Thank you

---

## [Decision Letter · Decision Letter 2]

12 Apr 2021

Trends and predictors of in-hospital mortality among babies with hypoxic ischaemic encephalopathy at a tertiary hospital in Nigeria: A retrospective cohort study

PONE-D-20-22771R2

Dear Dr. Ezenwa,

We’re pleased to inform you that your manuscript has been judged scientifically suitable for publication and will be formally accepted for publication once it meets all outstanding technical requirements.

Kind regards,

Ju Lee Oei

Academic Editor

PLOS ONE

Additional Editor Comments (optional):

Reviewers' comments:

Reviewer's Responses to Questions

**Comments to the Author**

1. If the authors have adequately addressed your comments raised in a previous round of review and you feel that this manuscript is now acceptable for publication, you may indicate that here to bypass the “Comments to the Author” section, enter your conflict of interest statement in the “Confidential to Editor” section, and submit your "Accept" recommendation.

Reviewer #3: All comments have been addressed

2. Is the manuscript technically sound, and do the data support the conclusions?

Reviewer #3: Yes

3. Has the statistical analysis been performed appropriately and rigorously? 

Reviewer #3: Yes

4. Have the authors made all data underlying the findings in their manuscript fully available?

Reviewer #3: Yes

5. Is the manuscript presented in an intelligible fashion and written in standard English?

Reviewer #3: Yes

6. Review Comments to the Author

Reviewer #3: I find this second revision has been thorough and addresses my major concerns well. The authors have fulfilled all the editing requirements.

7. PLOS authors have the option to publish the peer review history of their article (what does this mean?). If published, this will include your full peer review and any attached files.

Reviewer #3: No

---

## [Editor Report · Acceptance letter]

14 Apr 2021

PONE-D-20-22771R2 

Trends and predictors of in-hospital mortality among babies with hypoxic ischaemic encephalopathy at a tertiary hospital in Nigeria: a retrospective cohort study 

Dear Dr. Ezenwa:

I'm pleased to inform you that your manuscript has been deemed suitable for publication in PLOS ONE. Congratulations! Your manuscript is now with our production department. 

Kind regards, 

on behalf of

Dr. Ju Lee Oei 

Academic Editor

PLOS ONE